# Neural substrates of parallel devaluation-sensitive and devaluation-insensitive Pavlovian learning in humans

Eva R. Pool [1,2] ✉, Wolfgang M. Pauli[2,3], Logan Cross[4,5] & John P. O'Doherty [2,3]

We aim to differentiate the brain regions involved in the learning and encoding of Pavlovian associations sensitive to changes in outcome value from those that are not sensitive to such changes by combining a learning task with outcome devaluation, eye-tracking, and functional magnetic resonance imaging in humans. Contrary to theoretical expectation, voxels correlating with reward prediction errors in the ventral striatum and subgenual cingulate appear to be sensitive to devaluation. Moreover, regions encoding state prediction errors appear to be devaluation insensitive. We can also distinguish regions encoding predictions about outcome taste identity from predictions about expected spatial location. Regions encoding predictions about taste identity seem devaluation sensitive while those encoding predictions about an outcome's spatial location seem devaluation insensitive. These findings suggest the existence of multiple and distinct associative mechanisms in the brain and help identify putative neural correlates for the parallel expression of both devaluation sensitive and insensitive conditioned behaviors.

Pavlovian learning is one of the simplest and most fundamental forms of learning, whereby an initially neutral stimulus (conditioned stimulus or CS; e.g., a metronome sound) acquires or changes value by being associated with an affectively significant outcome (e.g., food)[1–3]. This form of associative learning exerts a profound influence on behavior[4,5], cognition[6,7], and mental health[8–10]. Despite being extensively studied across animals and humans, the neurocomputational mechanisms involved in Pavlovian learning appear to be more elaborate than previously conceived[11–16].

Value learning signals during Pavlovian conditioning have been extensively characterized in the brain. Reward prediction errors – a learning mechanism through which the CS becomes endowed with an outcome's affective value[17–20] – have been shown to correlate with dopaminergic activity in the midbrain[21], as well as blood oxygenation level dependent (BOLD) responses in the ventral striatum[22] and midbrain[23]. Moreover, the acquisition of affective conditioned responses appears to involve frontomedial structures such as the ventromedial prefrontal cortex (vmPFC)[24] and subgenual anterior cingulate cortex (sgACC)[25,26]. Lesion studies in monkeys[26] and humans[24] suggest that these structures are critical for a CS to trigger affective conditioned responses, reflected either in pupil dilation[26] or in skin conductance[24].

In the last decade, a growing number of studies have found evidence for other kinds of learning signals. Specifically, neural signals associated with model-based representations, or cognitive maps, have been identified during Pavlovian learning[5,11,13,27–29]. A key learning signal suggested to be involved in the building of a cognitive map is the state prediction error. This prediction error quantifies how unexpected a particular perceptual state is given the previous state, independently of its affective value and is implicated in the acquisition of a state–space transition model. State prediction errors have been reported in the lateral prefrontal cortex (PFC)[30], lateral orbitofrontal cortex (OFC), anterior insula, superior frontal gyrus (SFG)[31,32], and the intraparietal cortex[5,30]. It has been shown that reward prediction errors

[1]Swiss Center for Affective Sciences, University of Geneva, Geneva, Switzerland. [2]Division of Humanities and Social Sciences, California Institute of Technology, Pasadena, CA, USA. [3]Computation and Neural Systems Program, California Institute of Technology, Pasadena, CA, USA. [4]Division of Biology and Biological Engineering, California Institute of Technology, Pasadena, CA, USA. [5]Department of Computer Science, Stanford University, Palo Alto, CA, USA. ✉e-mail: eva.pool@unige.ch

in the ventral striatum exist alongside state prediction errors in other structures such as the lateral OFC[32]. The OFC has been implicated in the representation of states and cognitive maps in a large corpus of studies[33–37] and increasing evidence implicates OFC model-based representations in Pavlovian learning, including representations of outcome identity, an outcome's sensory features, sensory feature changes, and the acquisition of stimulus–stimulus associations underlying the construction of cognitive maps[11,27–29]. Interestingly, model-based learning signals during Pavlovian conditioning have also been found in brain regions typically involved in value learning such as the striatum[11], amygdala[4], and the dopaminergic midbrain – with activity coding for the updating of expectations about perceptual attributes of the outcome[28,38,39].

The co-existence of distinct neural signals related to an outcome's affective value on one hand and the perceptual properties of an outcome on the other, are broadly consistent with theoretical models postulating that Pavlovian learning is not a unitary process, but rather involves several parallel associations between the CS and multiple attributes of the outcome[40,41]. There has been a long-standing conceptualization of multiple and parallel conditioned responses to a given CS[42–44], but only recently have these classes of behavioral responses and their underlying neural learning signals been investigated in humans[16,45]. Strikingly, these parallel behavioral responses to a given CS diverge in their sensitivity to changes in outcome value, leading to the expression of conditioned responses—such as increased pupil dilation—that flexibly adapt to the updated value of an outcome, and others that persist unchanged, despite the outcome being devalued within the same individual[16].

Devaluation insensitive behaviors are often suggested to rely on brain signals approximated through model-free reinforcement learning algorithms that use reward prediction errors to make predictions based on cached values[14,46]. A key empirical test of this hypothesis as applied to Pavlovian conditioning would be whether brain regions correlating with model-free reinforcement learning based on reward prediction errors are sensitive to changes in outcome value. If brain regions sensitive to reward prediction errors are indeed insensitive to devaluation, this would provide evidence for the role of model-free reinforcement-learning in the acquisition of devaluation insensitive Pavlovian behaviors. On the other hand, if such reward prediction error signal coding brain regions are actually sensitive to outcome devaluation, this would suggest that Pavlovian reward prediction error-based learning is not model-free.

Within a model-based framework, some computations would be expected to be devaluation sensitive while others would not. Model-based predictive representations of expected-value should be devaluation sensitive by definition, as these representations are proposed to emerge by integrating knowledge of stimulus-stimulus associations with knowledge about current expected outcome value. On the other hand, internal representations of the cognitive model itself should not be sensitive to changes in outcome-value, for instance, information about where in the environment an outcome is expected to occur.

Here, we scanned human participants with fMRI while they performed a Pavlovian learning paradigm[16], in which they were asked to learn associations between various neutral images and videos of the delivery of a food outcome (see Fig. 1A). There were five images: one image was more often associated with the delivery of sweet food on the left side of the screen (CS+ left sweet); one image was more often associated with the delivery of the sweet outcome on the right side of the screen (CS+ right sweet); one image was more often associated with the delivery of salty food on the left side of the screen (CS+ left salty); one image was more often associated with the delivery of the salty outcome on the right side of the screen (CS+ right salty); and another image was more often associated with no outcome (CS−). We measured the pupil dilation at the CS onset as a conditioned response

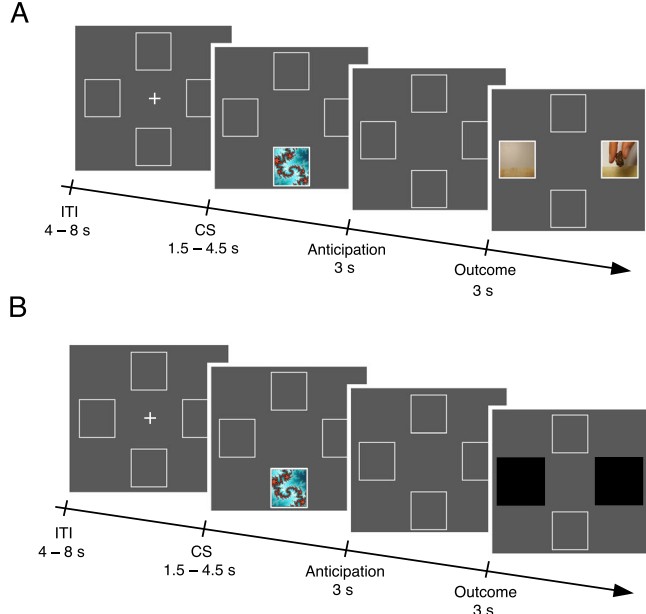

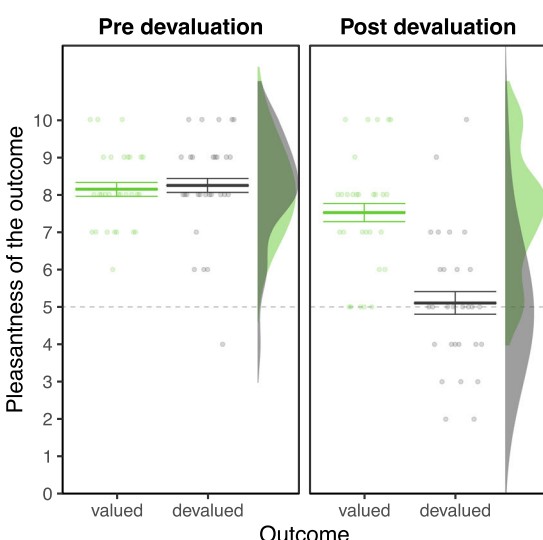

**Fig. 1 | Schematic representation of the experimental design. A** Illustration of the sequence of events in a trial during the acquisition phase administered before devaluation. At the beginning of each trial a conditioned stimulus (CS) was presented randomly in the upper or lower portion of the screen for 1.5–4.5 s (uniformly distributed). After an anticipation screen of 3 s, a video showing the snack delivery appeared either to the right or the left side of the screen for 3 s. Participants were asked to detect the location of the video of the snack delivery as rapidly as possible. The intertrial interval (ITI) lasted for 4–8 s (uniformly distributed). At the end of each run, participants received the actual snacks delivered during the task and were allowed to eat them. **B** Illustration of the sequence of events in a trial during the test phase administered after devaluation. All aspects were identical to the acquisition phase with the exception that the outcome delivery happened behind two black patches. **C** Manipulation check of the outcome devaluation procedure. Mean pleasantness ratings of the snack that was devalued through the selective satiation procedure (devalued pleasantness) and the snack that was not (valued pleasantness). Error bars indicate the within--participant s.e.m. N = 29 participants.

reflecting affective value[11,13,16] and the anticipatory gaze direction (left vs. right) as a conditioned response reflecting a specific perceptual representation of the outcome (i.e., its spatial location)[16]. We used these conditioned responses to fit a model that learns through reward prediction errors−tracking changes in affective value independently of the perceptual attributes of the outcome[18]−and a model that learns through state prediction errors−tracking how unexpected a particular perceptual outcome state is given the previous state independently of its affective value[5,30].

We identified brain regions involved in learning associations between the CS and an outcome's affective value as well as other attributes of an outcome such as its perceptual features, by correlating the BOLD signal with trial-by-trial reward- and state- prediction errors. We then tested the sensitivity of these identified regions to outcome devaluation. State prediction errors were found to carry information concerning predictions about two perceptual attributes of an outcome: its taste identity (sweet or salty) and its spatial localization (left or right). Therefore, to further investigate the representations of predictions about outcome attributes and their sensitivity to outcome devaluation, we performed a supplementary analysis. We implemented a multivoxel pattern analysis (MVPA) on the BOLD responses to the CS onset. We decoded predicted outcome taste identity by training a classifier to discriminate between the CS+ sweet and the CS+ salty associated with the outcome delivery to the left side and then tested its ability to discriminate between the CS+ sweet and the CS+ salty associated with the outcome delivery to the right side. Following the same logic, we decoded predicted outcome delivery location by training a classifier to discriminate between the CS+ left and the CS+ right associated with the sweet outcome and then tested its ability to discriminate between the CS+ left and the CS+ right associated with the salty outcome.

Using this approach, we aimed to test for the extent to which brain regions involved in implementing different learned associations in Pavlovian conditioning are sensitive to changes in outcome value. We further aimed to directly test for the applicability of the distinction between model-based and model-free reinforcement learning as a means of explaining differences in devaluation sensitivity across these different Pavlovian associations.

## Results

### Behavioral results

**Pavlovian learning.** During the acquisition phase, we tested whether pupil dilation and anticipatory gaze direction reflect patterns of distinct classes of Pavlovian response as in Zhang et al.'s study[45] and our previous study[16]. We expected pupil dilation to follow a value pattern (all CSs+ different from CS-) and gaze direction to follow a lateralized pattern (larger dwell time for CSs+ left compared to CSs+ right and the CS- on the left side of the screen; larger dwell time for CSs+ right compared to CSs+ left and the CS- on the right side of the screen).

**Pupil dilation.** As expected, a planned contrast analysis on the CS condition (CSs+ left, CSs+ right, CS- with the following weights: +0.5, +0.5, −1) revealed that the pupil was less constricted for CSs+ left and CSs+ right compared to CS- ($\beta = −0.030$, $SE = 0.011$, 95% CI = [− 0.053, − 0.007], $p = 0.016$, $BF_{10} = 3.68$; see Fig. 2A).

**Anticipatory gaze direction.** The first planned contrast analysis on the CS condition (CSs+ left, CSs+ right, CS- with the following

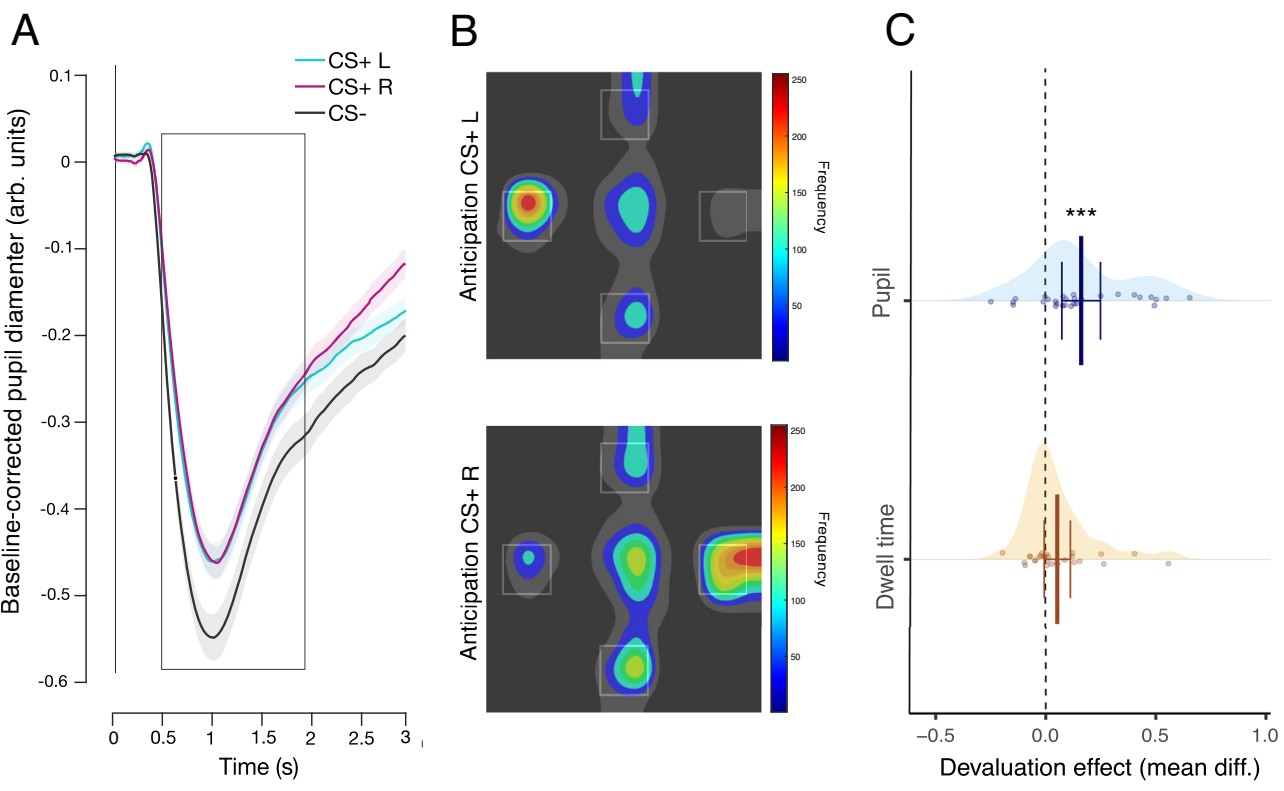

**Fig. 2 | Effects of Pavlovian conditioning and outcome devaluation on eye behavior. A** Averaged pupil response over time aligned to the conditioned stimulus (CS) onset for the CSs predicting either the delivery of a snack to the left (CS+ L), the delivery of a snack to the right (CS+ R) or no snack delivery (CS-). Shaded areas indicate the within--participant s.e.m. **B** Heatmaps of the fixation patterns during the anticipation screen (normalized frequency), after the offset of CS+ L and of the CS+ R. **C** Devaluation effect calculated as the mean difference of the devaluation induced change for the CS valued and the CS devalued (post[valued− devalued]−pre[valued−devalued]) in the pupil response (CS- corrected) and in the dwell time of the anticipatory gaze direction (CS- corrected). Error bars indicate 95% confidence interval adapted for within participants design. Statistical significance was determined by the interaction term (session: pre or post devaluation × CS: value or devalued) in a linear mixed-effects model. Asterisks indicate statistically significant differences ($\beta = 0.040$, $SE = 0.008$, 95% CI = [0.023, 0.057], $p < 0.001$, $BF_{10} = 44.77$). $N = 29$ participants.

weights: +1,-0.5, -0.5), revealed an increased dwell time in the left region of interest (ROI) after the perception of CSs+ left compared to CSs+ right and CS-($\beta = -0.072$, $SE = 0.019$, 95% CI = [$-0.110$, $-0.033$], $p = 0.0010$, $BF_{10} = 85.07$; see Fig. 2B). The second planned contrast analysis on the CS condition (CSs+ left, CSs+ right, CS- with the following weights: -0.5, +1, -0.5), revealed an increased dwell time in the right ROI after the perception of CSs+ right compared to CSs+ left and CS- ($\beta = -0.079$, $SE = 0.020$, 95% CI = [$-0.118$, $-0.039$], $p = 0.0005$, $BF_{10} = 53.18$; see Fig. 2B).

**Reaction times.** We tested whether participants' reaction times to the outcome delivery were influenced by Pavlovian predictions about (a) the outcome lateralization (i.e., left or right) and (b) the taste identity of the outcome (i.e., sweet or salty). For outcome lateralization, results showed that participants had a significantly longer reaction time when the side of the outcome (but not the identity) was different than the one most often predicted by the CS compared to when it was the same (e.g., *unexpected side effect*; $\beta = -0.054$, $SE = 0.012$, 95% CI = [$-0.078$, $-0.0306$], $p < 0.001$, $BF_{10} = 270.42$). For outcome identity, we did not find statistically significant effects, although descriptively participants showed longer reaction times when the identity of the outcome (but not the side) was different than the one most often predicted by the CS compared to when it was the same (e.g., *unexpected identity effect*; $\beta = -0.017$, $SE = 0.010$, 95% CI = [$-0.037$, $0.002$], $p = 0.101$, $BF_{10} = 0.579$).

**Outcome devaluation.** A statistically significant interaction between session (pre- or post-satiation) and outcome (valued or devalued) showed that the outcome devaluation procedure decreased the pleasantness of the devalued food outcome in comparison to the valued food outcome ($\beta = 0.629$, $SE = 0.113$, 95% CI = [$0.406$, $0.869$], $p > 0.001$, $BF_{10} = 434.23$; see Fig. 1C and Supplementary Fig. 4).

**Outcome devaluation induced changes.** To test for sensitivity to outcome value, we compared the change induced by devaluation in the differential conditioned responses (i.e., $CS+ - CS-$) to the still valued CS+ to the change induced by devaluation to the devalued CS+ in a 2 (session: pre- or post-satiation) by 2 (CS: valued or devalued) interaction. We expected the conditioned pupil response to adapt more readily to outcome devaluation than the conditioned anticipatory gaze direction.

**Pupil dilation.** We averaged the pupil response of the CSs associated with the valued outcome and the CSs associated with the devalued outcome and corrected it by subtracting the average pupil dilation during the CS-. We did this operation at two time points: the last run before satiation and the test run. A statistically significant interaction between session and CS showed that the decrease in pupil dilation induced by satiation was larger for the CSs associated with the devalued outcome than the CSs associated with the valued outcome ($\beta = 0.040$, $SE = 0.008$, 95% CI = [$0.023$, $0.057$], $p < 0.0001$, $BF_{10} = 44.77$; see Fig. 2C and Supplementary Fig. 5A).

**Anticipatory gaze direction.** We averaged dwell time allocated to the congruent region of interest (ROI) for all the CSs+ (dwell time in the right ROI after CSs+ right and dwell time in the left ROI after CSs+ left) for the CSs associated with the devalued outcome (CS devalued) and the CSs associated with the valued outcome (CS valued) and corrected it by subtracting the averaged dwell time during the CS- over both ROI. We did this operation at two time points: the last session before satiation and the test session. We did not find evidence for an interaction between session and CS ($\beta = 0.013$, $SE = 0.007$, 95% CI = [$-0.0006$, $0.027$], $p = 0.0710$, $BF_{10} = 0.262$; see Fig. 2C and Supplementary Fig. 5B).

**Reaction times.** We also measured reaction times taken to guess which video was being displayed behind the black patches during the test session following the CS associated with the valued outcome and the devalued outcome. We did not find a statistically significant

difference between the CS valued and the CS devalued conditions ($\beta = 0.006$, $SE = 0.008$; 95% CI = [$-0.009$, $0.021$], $p = 0.459$, $BF_{10} = 0.440$).

## fMRI Results

**Parallel Pavlovian predictions about affective value and perceptual attributes of the outcome.** To identify the brain ROIs separately involved in implementing Pavlovian predictions about the affective value and perceptual attributes of the outcome, respectively, we derived trial-by-trial prediction errors during the first two runs from two models: one learning through reward prediction errors—tracking changes in affective value, independently of the perceptual attributes of the outcome itself; and the other learning through state prediction errors—tracking how unexpected a particular perceptual outcome state is independently of its affective value. We then tested for the sensitivity to devaluation of the ROIs identified with these two models.

**Reward prediction errors.** We tested the devaluation sensitivity of the brain regions involved in reward prediction error coding. To do so, we defined ROIs by extracting the contrast correlating with the trial-by-trial reward prediction errors. We focused on three ROIs identified by this contrast: one ROI covering parts of the ventral striatum and of the sgACC (VS / sgACC), a second ROI covering parts of the midbrain, and a third ROI covering parts of the vmPFC (see Fig. 3A and Table 1).

To test for devaluation effects inside these ROIs, we compared activity while participants expected a valued versus a devalued outcome, during the run after the devaluation procedure. We also used pseudo-extinction, whereby the visual presentation of the outcomes was obscured behind two black patch covers present at the time of the outcome delivery. Pseudo-extinction is a crucial manipulation that prevents rapid relearning of a CS's expected value via the newly devalued outcome. Thus, this procedure allows predictive representations linked to the incentive value of the predicted outcome to be dissociated from those associated with outcome-value insensitive representations. We observed a statistically significant devaluation effect in the VS / sgACC ROI ($\beta = -0.149$, $SE = 0.057$, 95% CI = [$-0.267$, $-0.030$], $p = 0.0157$, $BF_{10} = 2.98$; see Fig. 3B and Supplementary Fig. 6), which survived correction for multiple comparisons across ROIs. We did not find statistical evidence for a devaluation effect in the midbrain ROI ($\beta = -0.037$, $SE = 0.054$, 95% CI = [$-0.149$, $0.074$], $p = 0.498$, $BF_{10} = 0.230$; see Fig. 3B) and the vmPFC ROI ($\beta = -0.190$, $SE = 0.142$, 95% CI = [$-0.481$, $0.099$], $p = 0.189$, $BF_{10} = 0.390$; see Fig. 3B and Supplementary Fig. 6).

**State prediction errors.** We next tested the devaluation sensitivity of the brain regions putatively involved in model-based learning, such as when forming stimulus–stimulus associations between stimuli and an outcome's perceptual features. To do so, we defined ROIs by extracting the contrast correlating with the trial-by-trial state prediction errors, which tracked how unexpected a particular outcome state is, given the previous state. We focused on four ROIs identified from this contrast: one covering parts of the lateral orbitofrontal cortex and anterior insula (OFC), a second covering parts of the middle frontal gyrus and inferior frontal gyrus (MFG), a third covering parts of the superior frontal gyrus (SFG), and a fourth covering parts of the midbrain (see Fig. 3C and Table 2)

We did not find evidence for a statistically significant effect of devaluation in the MFG ROI ($\beta = -0.124$, $SE = 0.174$, 95% CI = [$-0.482$, $0.233$], $p = 0.480$, $BF_{10} = 0.298$; see Fig. 3D and Sup. Fig. 7), the SFG ROI ($\beta = -0.031$, $SE = 0.259$, 95% CI = [$-0.455$, $0.392$], $p = 0.878$, $BF_{10} = 0.258$; see Fig. 3D and Sup. Fig. 7), the OFC ROI ($\beta = -0.039$, $SE = 0.141$, 95% CI = [$-0.320$, $0.241$], $p = 0.781$, $BF_{10} = 0.281$; see Fig. 3D and Supplementary Fig. 7), or the midbrain ROI ($\beta = 0.0526$, $SE = 0.114$, 95% CI = [$-0.182$, $0.287$], $p = 0.649$, $BF_{10} = 0.208$; see Fig. 3D and Supplementary Fig. 7)

State prediction errors could potentially be involved in mediating learning about two different perceptual attributes of an outcome: a

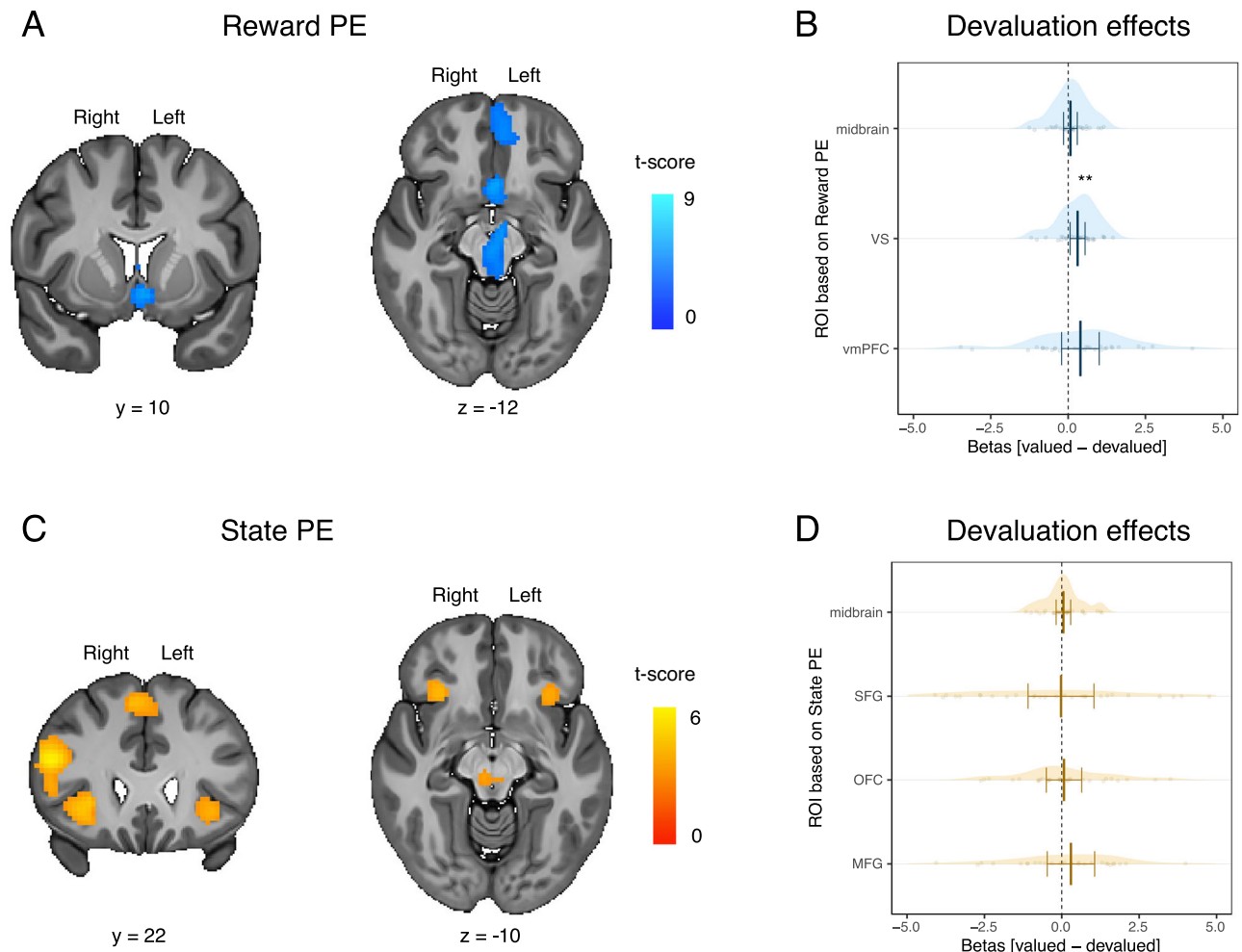

**Fig. 3 | Reward and state prediction errors and sensitivity to outcome devaluation. A** Brain regions correlating with reward prediction error (Reward PE). Sensitivity to outcome devaluation was estimated by calculating the mean difference between the betas for the valued contrast - betas for devalued contrast in the regions of interest (ROI). **B** Sensitivity to outcome devaluation in the midbrain ROI the ventral striatum / sgACC ROI (VS), the ventromedial prefrontal cortex ROI (vmPFC). **C** Brain regions correlating with state prediction error (State PE). **D** Sensitivity to outcome devaluation in the midbrain ROI, the superior frontal gyrus ROI (SFG), the bilateral orbitofrontal/anterior insula ROI (OFC), the middle prefrontal gyrus/inferior frontal gyrus ROI (MFG). The valued contrast was defined as the difference in the

BOLD signal at the outcome delivery (displayed behind two black patches) after the perception of the positive conditioned stimulus (CS+) valued versus the negative conditioned stimulus (CS-). The devalued contrast was defined as the difference in the BOLD signal at the outcome delivery (displayed behind two black patches) after the perception of the CS+ devalued versus the CS-. Error bars indicate 95% confidence interval adapted for within participants design. Statistical significance was determined by the effect of the outcome value (value or devalued) in a linear mixed model. Asterisks indicate the statistically significant difference that survives correction for multiple comparisons across ROI ($\beta = -0.149$, $SE = 0.057$, 95% CI = [$-0.267$, $-0.030$], $p = 0.0157$, $BF_{10} = 2.98$). $N = 29$ participants.

stimulus could be unexpected because of a violation in the expected taste identity of the outcome (sweet or salty) or because of an unexpected arrival of the outcome in a particular spatial location (left or right side). To test to what extent these two aspects are reflected in the state prediction error brain signals, we extracted the $\beta$ effect of the state prediction error from the state prediction error ROIs and averaged this across the different ROIs. Then we correlated the averaged $\beta$ effect of the state prediction error against the unexpected side effect and the unexpected identity effect as measured with reaction times during the two runs of the acquisition phase. More precisely, to compute a reaction time index reflecting the unexpected side effect, we only used the trials where the identity was the one most often predicted by the CS but the side of the outcome varied. We subtracted the average reaction time on trials where the side of the outcome was the same as the one most often predicted by the CS from the average reaction time on trials where the side was different from the one most often predicted by the CS. To compute a reaction time index reflecting the unexpected identity effect, we only used the trials where the side

was the one most often predicted by the CS but the identity of the outcome varied. We subtracted the average reaction time in trials where the identity was the same as the one most often predicted by the CS from the average reaction time on trials where the identity was different from the one most often predicted by the CS.

We found that the magnitude of state prediction errors was associated with the magnitude of the unexpected identity effect in reaction times ($\beta = 1.956$, $SE = 0.830$, 95% CI = [0.339, 3.583], $p = 0.026$, $BF_{10} = 3.05$; see Fig. 4A) and with the magnitude of the unexpected side effect in reaction times ($\beta = 1.883$, $SE = 0.671$, 95% CI = [0.566, 3.199], $p = 0.009$, $BF_{10} = 7.30$; see Fig. 4A). As a control, we correlated these behavioral indexes with the $\beta$ effect of reward prediction errors. We did not find conclusive evidence that reward prediction errors are associated with the magnitude of either the unexpected taste identity effect ($\beta = 0.966$, $SE = 0.520$, 95% CI = [$-0.053$, 1.986], $p = 0.066$, $BF_{10} = 1.40$) or the unexpected side effect ($\beta = 0.587$, $SE = 0.441$, 95% CI = [$-0.277$, 1.452], $p = 0.195$, $BF_{10} = 0.816$) in the reaction times (see Fig. 4B).

**Table 1 | Summary of the results for the BOLD activations correlating with reward prediction errors**

| Region | Laterality | Extent | β (SE) | 95% CI | BF₁₀ | Coordinates | | |
|---|---|---|---|---|---|---|---|---|
| | | | | | | x | y | z |
| Precentral gyrus / Postcentral gyrus | R | 908 | | | | −55 | −21 | 36 |
| Precentral gyrus / Postcentral gyrus | L | 1506 | | | | −32 | −24 | 50 |
| Midbrain ★ | R | 242 | 0.428 (0.079) | [0.265, 0.591] | 2226 | 0 | −18 | −7 |
| VS / sgACC ★ | L | 142 | 0.350 (0.060) | [0.227, 0.473] | 6804 | −2 | 6 | −7 |
| vmPFC ★ | L | 140 | 0.545 (0.126) | [0.287, 0.804] | 156.65 | −8 | 52 | −14 |

Thresholding $t(28) > 3.41$, $p < 0.001$, and minimum cluster level simulation extent for multiple comparisons correction at $p < 0.05 = 101$. Coordinates are expressed in the Montreal Neurological Institute (MNI) space in the left-right, anterior-posterior, and inferior-superior dimensions, respectively.
★ indicates activations used to define ROIs. $N = 29$ participants.
*vmPFC* ventromedial prefrontal cortex, *VS* ventral striatum, *sgACC* subgenual anterior cingulate cortex.

**Table 2 | Summary of the results for the BOLD activations correlating with state prediction errors**

| Region | Laterality | Extent | β (SE) | 95% CI | BF₁₀ | Peak Coordinates | | |
|---|---|---|---|---|---|---|---|---|
| | | | | | | x | y | z |
| MFG / IFG ★ | R | 748 | 0.562 (0.119) | [0.317, 0.808] | 390.29 | 50 | 29 | 20 |
| OFC / anterior insula ★ | R | | 0.425 (0.103) | [0.224, 0.676] | 158.60 | 30 | 24 | −2 |
| Midbrain ★ | R | 170 | 0.181 (0.036) | [0.106, 0.255] | 786.36 | 8 | −28 | −14 |
| SFG ★ | R | 288 | 0.568 (0.132) | [0.297, 0.840] | 146.29 | 5 | 22 | 50 |
| OFC / anterior insula ★ | L | 158 | 0.426 (0.103) | [0.215, 0.637] | 100.59 | −35 | 24 | −4 |

Thresholding $t(28) > 3.41$, $p < 0.001$, and minimum cluster level simulation extent for multiple comparisons correction at $p < 0.05 = 101$. Coordinates are expressed in the Montreal Neurological Institute (MNI) space in the left-right, anterior-posterior, and inferior-superior dimensions, respectively.
★ indicates activations used to define ROIs. $N = 29$ participants.
*MFG* Medial frontal gyrus, *IFG* Inferior frontal gyrus, *OFC* orbitofrontal cortex, *SFG* superior frontal gyrus.

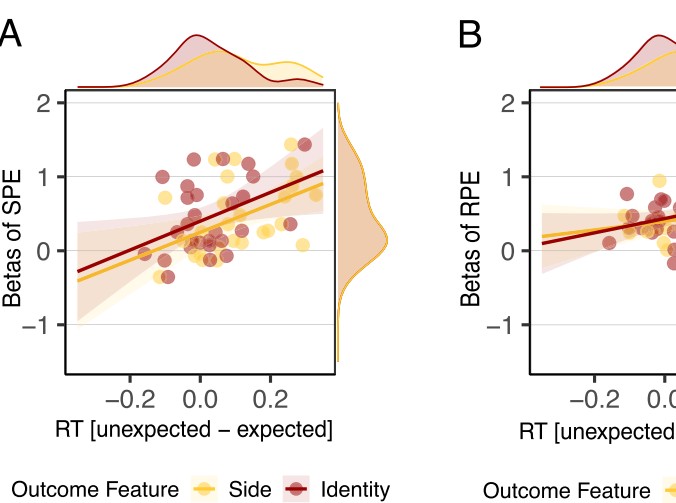

**Fig. 4 | Correlation between state and reward prediction errors and unexpected effects measured with reaction times during the acquisition phase.**
**A** Correlation between the magnitude of the state prediction error (SPE) effect and difference in the reaction times (RT) to detect the outcome when the side was unexpected vs expected (in yellow) and when the taste identity was unexpected vs expected (in red). **B** Correlation between the magnitude of the reward prediction error (RPE) effect and difference in the reaction times to detect the outcome when the side was unexpected vs expected (in yellow) and when the taste identity was unexpected vs expected (in red). Shaded area represents 95% confidence interval. $N = 29$ participants.

**Pavlovian predictions about spatial location and taste identity attributes of the outcome.** In a supplementary analysis, we identified two distinct sets of brain regions involved in encoding predictions about outcome taste identity and predictions about outcome delivery location, respectively, using a multivariate pattern analysis on the BOLD responses to the CSs. To identify voxels encoding predictions about an outcome's taste attributes, we trained a classifier to discriminate between the CSs+ sweet and CSs+ salty cues associated with outcome delivery to one side of the screen (e.g., CS+ left sweet and CS+

left salty) and then tested for its ability to discriminate between the CSs+ sweet and CSs+ salty associated with outcome delivery on the other side of the screen (e.g., CS+ right sweet and CS+ right salty). To identify voxels encoding predictions about the spatial location (side) of outcome delivery, we followed the same logic: we trained the classifier to decode between the CSs+ left and CSs+ right associated with a specific taste outcome (e.g., CS+ left sweet and CS+ right sweet) and then tested its ability to discriminate between the CSs+ left and right associated with the other taste outcome (e.g., CS+ left salty and CS+

right salty). We then tested the sensitivity to outcome devaluation of these two different sets of ROIs, thereby enabling us to determine whether different kinds of predictive representations about perceptual outcome attributes might have differential sensitivity to changes in outcome value. Please note that to define the ROIs for their use in the subsequent outcome devaluation test, we used a liberal threshold of $p < 0.005$ uncorrected (with an extent threshold of 100) on voxels identified from the two MVPA analyses.

**Representations of predicted taste identity.** We defined our first set of ROIs based on decoding the predicted taste identity (sweet or salty) of the outcome during the time of the CS onset. We focused on four ROIs identified with this approach: one ROI in the right inferior frontal gyrus (IFG), a second ROI covering the the superior temporal lobule and the right intraparietal sulcus (IPS), a third ROI covering the left paracentral lobule and post central gyrus (PCL), and a forth ROI covering the post central gyrus (PCG; see Table 3 and Supplementary Fig. 2).

We then tested for devaluation sensitivity inside these ROIs, by comparing devaluation induced changes in the BOLD signal between the valued and devalued conditions. We found a statistically significant effect of devaluation sensitivity in the left PCG: $\beta = -0,187$, $SE = 0.061$, 95% CI $= [-0.313, -0.061]$, $p = 0.005$, $BF_{10} = 4.73$, which survived correction for multiple comparisons across ROIs. We found an effect pointing in the same direction in the left PCL ($\beta = -0.0713$, $SE = 0.030$, 95% CI $= [-0.134, -0.008]$, $p = 0.027$, $BF_{10} = 0.937$) and the right IPS ($\beta = -0.118$, $SE = 0.046$, 95% CI $= [-0.213, -0.022]$, $p = 0.017$, $BF_{10} = 1.59$), but these effects did not survive correction for multiple comparisons across ROIs. We did not find any statistically significant effect in the right IFG ($\beta = -0.064$, $SE = 0.083$, 95% CI $= [-0.235, 0.106]$, $p = 0.445$, $BF_{10} = 0.155$; see Fig. 5 and Supplementary Fig. 8).

**Representations of the predicted spatial location of an outcome (side).** We defined our second set of ROIs based on decoding the predicted spatial location (left or right) of the outcome during the time of onset of the CS. We focused on four ROIs identified with this approach: one ROI covering left and right portions of the cuneal and calcarine cortex (Cuneus), a second ROI covering the right superior temporal lobule and intraparietal sulcus (IPS), a third ROI covering parts of the left and right supra marginal gyrus (SMG), and a forth ROI covering the parts of the right middle temporal gyrus and the lateral occipital cortex (LOC; see Table 4 and Supplementary Fig. 2).

Then, we tested for devaluation sensitivity inside these ROIs by comparing devaluation induced changes in the BOLD signal between the valued and devalued conditions. We did not find any statistical evidence for devaluation sensitivity in the ROIs defined based on the predicted outcome spatial location (Cuneus: $\beta = -0.096$, $SE = 0.051$, 95% CI $= [-0.203, 0.009]$, $p = 0.073$, $BF_{10} = 0.640$; right IPS: $\beta = -0.036$, $SE = 0.034$, 95% CI $= [-0.108, 0.034]$, $p = 0.300$, $BF_{10} = 1.001$; right LOC: $\beta = -0.080$, $SE = 0.061$, 95% CI $= [-0.206, 0.045]$, $p = 0.201$, $BF_{10} = 0.357$; SMG: $\beta = -0.089$, $SE = 0.062$, 95% CI $= [-0.216, 0.0379]$, $p = 0.161$, $BF_{10} = 0.631$; see Fig. 5 and Supplementary Fig. 9).

## Discussion

This study aimed to investigate how the brain encodes parallel associations between a CS and multiple attributes of an outcome, and to address which of these associations are sensitive to outcome devaluation. To this end, we combined an appetitive Pavlovian learning task with eye-tracking and fMRI measures and an outcome devaluation manipulation. We found evidence for parallel representations of outcome attribute predictions, relying on distinct brain regions that differ with respect to their sensitivity to outcome devaluation. Specifically, while a subset of regions involved in encoding reward prediction errors (the ventral striatum and sgACC) were found to be sensitive to devaluation, a different brain network involved in encoding state prediction errors appeared to be less sensitive to changes in outcome value. These distinct brain areas underlie different classes of conditioned responses such as pupil dilation—an indicator of affective value that flexibly adapts to outcome devaluation without the need to resample environmental contingencies—and approach tendencies in gaze behavior—a measure of perceptual properties of an outcome that appear to be less sensitive to changes in outcome value.

Learning processes underpinning predictions about an outcome's affective value associated with reward prediction errors were localized in medial brain regions such as vmPFC, sgACC, and ventral striatum; these regions have been typically implicated in value representation[47,48], value learning[22], and affective representations in appetitive Pavlovian learning reflected in conditioned responses measured with pupil dilation[26]. Learning processes underpinning stimulus–stimulus learning involving predictions about perceptual attributes of an outcome were by contrast found in more lateral brain regions such as the lateral PFC, the lateral OFC and the anterior insula, but also the SFG. These regions have been argued to play a role in model-based processes during value learning. Specifically, the lateral OFC has been implicated in the representation of states and cognitive maps in reinforcement learning and decision-making[33–36] and in the representation of perceptual attributes of outcomes in Pavlovian conditioning[11,29]. By comparison, activity in the dorsolateral PFC[30,31] and in the anterior insula and SFG[31] has been found to correlate with state prediction errors. We found evidence for the involvement of the midbrain in associative learning processes related to both perceptual and affective value attributes of the outcome. This result is congruent with seminal findings showing that dopaminergic activity in the midbrain codes reward prediction errors[21] and with recent findings showing that the midbrain might also be involved in prediction errors about the perceptual identity of a reward[28,38,39,49].

Interestingly, we found evidence for devaluation sensitivity in the ventral striatum and sgACC, which were part of the brain network involved in learning predictions about an outcome's affective value, but not in the brain regions involved in learning predictions about the perceptual properties of the outcome. The finding of flexible adaptation to changes in outcome value in ventral striatal regions in Pavlovian conditioning is congruent with previous findings in human fMRI

**Table 3 | Summary of the searchlight results for the decoding of predicted outcome taste identity**

| Region | Laterality | Extent | ACC (SE) | 95% CI | $BF_{10}$ | Coordinates | | |
|---|---|---|---|---|---|---|---|---|
| | | | | | | x | y | z |
| IFG ★ | R | 174 | 0.525 (0.007) | [0.510, 0.540] | 23.055 | 52 | 22 | 18 |
| IPS ★ | L | 179 | 0.534 (0.009) | [0.514, 0.553] | 31.317 | 38 | −38 | 56 |
| PCL ★ | L | 401 | 0.543 (0.009) | [0.524, 0.563] | 428.646 | −38 | 26 | 78 |
| PCG ★ | L | 113 | 0.531 (0.010) | [0.510, 0.552] | 8.522 | −62 | −8 | 36 |

Thresholding $p_{uncorr} > 0.005$, $k = 100$. Coordinates are expressed in the Montreal Neurological Institute (MNI) space in the left-right, anterior-posterior, and inferior-superior dimensions, respectively. ACC = classifier accuracy.

★ indicates activations used to define ROIs. $N = 29$ participants.

*IFG* inferior frontal gyrus, *IPS* intraparietal solcus and parts of the superior temporal lobule, *PCL* paracentral lobule and parts of the post central gyrus, *PCG* post central gyrus.

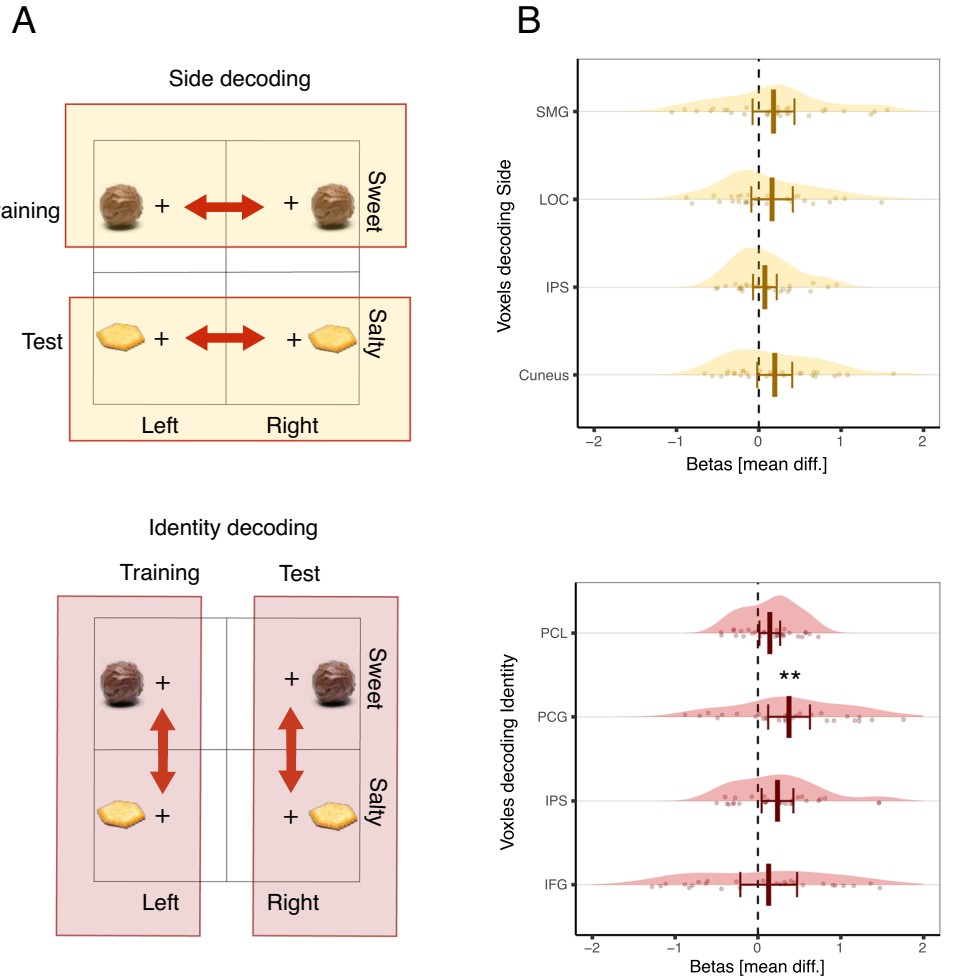

**Fig. 5 | Decoding predictions about spatial location (side) and taste identity representations of the outcome. A** Schematic of the decoding strategy. To decode the predicted location of the outcome, we trained a classifier to distinguish the left versus right side of outcome delivery at the onset of the CSs associated with the sweet outcome and then tested the classifier's ability to discriminate the left versus right side of the outcome delivery at the onset of the CSs associated with the salty outcome and vice-versa. To decode the predicted taste identity representation of the outcome, we trained a classifier to distinguish between the sweet versus salty outcome at the onset of the CSs associated with the outcome delivery to the left side and then tested its ability to discriminate the sweet versus salty outcome at the onset of the CSs associated with the outcome delivery to the right side of the screen and vice-versa. **B** Sensitivity to outcome devaluation was evaluated by calculating the mean difference between the differences in betas for the CS valued and the CS devalued pre- and post-devaluation (post[valued - devalued] - pre[valued - devalued]) in the regions of interest (ROI) identified with the MPVA analysis for decoding predicted side and predicted taste identity of the outcome. Error bars represent 95% confidence interval adjusted for within participants designs. Statistical significance was determined via a linear mixed-effects model. Double asterisks indicate the statistically significant difference that survives correction for multiple comparisons across ROIs ($\beta = -0,187$, $SE = 0.061$, 95% CI = [$-0.313$, $-0.061$], $p = 0.005$, $BF_{10} = 4.73$). $N = 29$ participants. IPS intraparietal sulcus, SMG supra marginal gyrus, LOC lateral occipital complex, IFG inferior frontal gyrus, PCL paracentral lobule, PCG post central gyrus.

**Table 4 | Summary of the searchlight results for the decoding of predicted outcome spatial location (side)**

| Region | Laterality | Extent | ACC (SE) | 95% CI | BF$_{10}$ | Coordinates | | |
|---|---|---|---|---|---|---|---|---|
| | | | | | | x | y | z |
| Cuneus ★ | L | 1972 | 0.593 (0.022) | [0.547, 0.639] | 105.898 | −21 | −81 | 8 |
| Outside atlas | | 111 | | | | −32 | −28 | 78 |
| IPS ★ | R | 156 | 0.527 (0.011) | [0.503, 0.551] | 2.081 | 20 | −51 | 63 |
| SMG ★ | R | 147 | 0.524 (0.009) | [0.505, 0.544] | 3.302 | 60 | −31 | 23 |
| SMG ★ | L | 123 | 0.527 (0.009) | [0.507, 0.547] | 5.827 | −65 | −44 | 26 |
| LOC ★ | R | 145 | 0.539 (0.0140) | [0.510, 0.567] | 4.635 | 52 | -66 | 3 |

Thresholding $p_{uncorr} > 0.005$, $k = 100$. Coordinates are expressed in the Montreal Neurological Institute (MNI) space in the left-right, anterior-posterior, and inferior-superior dimensions, respectively.
★ indicates activations used to define ROIs. $N = 29$ participants.
ACC classifier accuracy, IPS intraparietal sulcus, SMG supra marginal gyrus, LOC lateral occipital complex.

studies[50]. We did not find evidence for devaluation sensitivity in ROIs coding for state prediction error signals, even in the anterior insula and lateral OFC. However, we note that previous studies have found that parts of the lateral OFC are sensitive to devaluation procedures[50–52]. In the present study, we used a functional localizer wherein we likely identified a portion of the OFC that specifically encodes predictions about perceptual features and found that this portion of the lateral OFC is insensitive to devaluation. Thus, it is possible that multiple representations exist simultaneously in the lateral OFC, some of which are sensitive and some of which are insensitive to devaluation.

The results of the current study also align well with our prior behavioral findings[16]. In that previous study, by measuring different classes of Pavlovian responses with eye tracking and tracking their sensitivity to outcome devaluation, we found that conditioned pupil dilation reflects the outcome's affective value because it adapted to changes in outcome value. On the other hand, the tendency to adjust gaze direction toward the expected spatial location of the outcome – a key perceptual feature of the outcome – was found instead to be relatively insensitive to changes in outcome value. We also replicated these behavioral findings in the current study. These findings might appear to be at odds with a large portion of the traditional Pavlovian learning literature in which it is typically reported that responses based on sensory features of the outcome are indeed sensitive to outcome devaluation[53–56]. In our main analysis, we identified brain regions involved in learning affective value and perceptual attributes of an outcome by identifying regions correlating with reward and state prediction errors, respectively. However in the present study, we found that state prediction errors could play a role in facilitating learning about two different predictions about an outcome's perceptual attributes: its taste identity and its spatial localization. This possibility is supported by brain-behavior correlations showing that the magnitude of the BOLD signal in brain regions coding for state prediction errors is correlated with slower reaction times when an outcome was delivered in an unexpected spatial location and also when an unexpected taste identity was delivered following a given CS. Thus, these state prediction error coding regions are sensitive to two different perceptual features of an outcome.

To address the differential outcome sensitivity of predictions about outcome taste identity and outcome spatial location, respectively, we then performed a supplementary but more fine-grained MVPA analysis in which we identified distinct sets of regions encoding predictions about spatial location and taste identity. We found that these two distinct sets of regions responded in markedly different ways following outcome devaluation. While we did not find evidence for outcome devaluation sensitivity in voxels encoding predictions about the spatial location of the outcome, we found evidence for a flexible adaptation to the new outcome value in some of the voxels encoding predictions about outcome taste identity. This finding is consistent with previous findings from Howard et al.[57], reporting that brain regions decoding taste identity appear to be sensitive to outcome devaluation procedures. More importantly, these findings could reconcile the present findings with the classical animal literature showing that Pavlovian responses based on specific sensory representations of the outcome are sensitive to manipulations of outcome value[15,53–56]. Specifically, in the present study, we found that a subset of the representations involving predictions about those perceptual features relevant to outcome identity are sensitive to changes in outcome value, while predictive representations about other outcome features such as expected spatial location are not sensitive to changes in outcome value. Please note that this interpretation relies on a supplementary analysis carried out with a lenient threshold to define the regions of interest, therefore gathering further empirical data would be prudent before drawing more definitive conclusions. It is also important to emphasize that our devaluation sensitivity test relied on a univariate approach, which would not be sensitive to changes in outcome value representations manifesting via changes in multi-voxel patterns.

A complementary approach to investigate representations of outcome side and outcome identity would be to derive specific state prediction errors from models that only track the taste identity of the outcome (irrespective of the side of the delivery) and models that only track the side of the outcome delivery (irrespective of the taste identity of the outcome). In the present study, due to a limited amount trials available to test for these specific conditions in our design, we could not use this approach; but future studies could perform experiments designed to parse different aspects of the outcome based on different kinds of state prediction errors.

From a theoretical point of view, our findings support models of Pavlovian conditioning whereby Pavlovian learning is not a unitary process, but rather involves multiple and parallel forms of CS–outcome attribute associations[40,41], reflected at a behavioral level in multiple classes of Pavlovian conditioned responses[42,44,45]. Classical models of Pavlovian learning[42] distinguish between two classes of Pavlovian responses triggered in parallel by the same conditioned stimulus: preparatory responses influenced by the affective or motivational value of the outcome (e.g., heart rate) and consummatory responses influenced by the perceptual attributes of the outcome (e.g., chewing vs. liking for a liquid vs. solid food outcome). Other models[40,41] have extended this parallel learning model to multiple associations between the CS and different attributes of the outcome such as perceptual features, motivational, hedonic, and even temporal attributes.

We found a pattern of sensitivity to outcome devaluation in regions encoding reward and state prediction errors that largely contradicts the pattern that would be expected from a straightforward transposition to the Pavlovian domain of the model-based versus model-free dichotomy from instrumental conditioning[14,46]. Contrary to the predictions of that theory, some of the regions encoding reward prediction errors were sensitive to changes in outcome value. The reported sensitivity to outcome devaluation of regions involved in reward prediction error coding is contrary to theoretical predictions about reward prediction errors being purely model-free. Nonetheless, our findings are compatible with several recent findings in the animal literature supporting the idea that reward prediction errors carry model-based information[28,49,58–60] and that they show sensitivity to changes in outcome value[59]. In the present study, we found such devaluation sensitive signals in the ventral striatum and sgACC. Because BOLD fMRI signals are thought to reflect inputs into a region alongside intrinsic computations therein[61,62], the ventral striatum and sgACC responses we found in the present study could reflect at least in part, the effects of dopaminergic input. On the other hand, reward prediction errors in the midbrain were found to be insensitive to changes in outcome value. When taken together, these results could suggest the existence of both model-based and model-free reward-prediction error signals in parallel. The reward-prediction error codes found in the ventral striatum perhaps reflect a convergent influence of model-based signals on reward prediction error computations[63], while the signals observed in the midbrain itself could instead reflect a version of the reward prediction error signal that is clearly more model-free. Our findings suggest heterogeneity within reward-prediction error codes in the brain, supporting the possibility that reward-prediction errors could facilitate the acquisition of both devaluation sensitive and devaluation insensitive predictive value representations. The present findings add to a burgeoning literature suggesting that the computations underlying Pavlovian learning might be more complex than previously thought[14,15,64].

To conclude, our findings support the notion that multiple processes play a role in the construction of Pavlovian cognitive maps[40,41]. They further suggest that some of these processes focus on aspects that are less reactive to changes in outcome value and others that more readily react to changes in outcome value. The present study could

also provide new perspectives on problematic reward-seeking behaviors that characterize many psychological disorders such as substance use disorders, binge eating, and gambling. The persistent pursuit of outcomes that are no longer valued has been typically conceived as being controlled by stimulus–response mechanisms that do not rely on internal representations[65–67] in instrumental learning. Here, we found evidence suggesting that Pavlovian responses which persist despite changes in outcome value might actually rely on internal representations of certain perceptual features of the outcome. The potential over-representation of some devaluation insensitive outcome attributes during Pavlovian learning could therefore be an additional candidate mechanism behind pathological situations where outcomes that are no longer valuable are nevertheless assigned high behavioral priority.

## Methods

### Participants

Thirty healthy volunteers participated in this fMRI study. One participant had to be excluded from the analysis because of a hardware failure during data acquisition. The remaining sample was composed of 29 participants (11 females; 18 males) with a mean age of 24 years (SD = 8.4 years). Gender and age were self-reported. Written informed consent was obtained from all participants, according to a protocol approved by the Human Subjects Protection committee of the California Institute of Technology (Pasadena, CA). Participants received $50 compensation for their participation in this study. The sample size was determined based on the smallest devaluation effect in the pupil we found in a series of previous studies using similar behavioral paradigms[16]. Participants were prescreened to ensure they were not dieting and they were asked not to eat for at least 6 h before the experimental session (they were allowed to drink water).

### Materials

**Stimuli.** All stimuli were identical to Experiment 2 reported in[16]. The Pavlovian cues consisted of five neutral fractal images. The reward outcome consisted of a 3 s long video of the experimenter's hand delivering the participant's favorite snack into a small bag. At the end of each run, participants received the bag containing the snacks they had collected during the task, to consume. The correspondence between the amount of food consumed at the end of each session was not identical (i.e., 1 video:1 piece of snack) but proportional. The proportion varied from 1:2 to 1:6 according to the amount of calories per individual piece of the snack selected by the participant. All stimuli were displayed on a computer screen with a visual angle of 6° using Psychtoolbox 3.0 (http://psychtoolbox.org/), a visual interface implemented on Matlab (version 8.6; The Mathworks Inc.).

**Pupil dilation and gaze direction.** Pupil dilation and gaze direction were used to reflect two classes of Pavlovian responses: The pupil dilation on cue presentation was used as an index reflecting a response based on the value representation of the outcome; the anticipatory gaze direction was used as an index reflecting a response based on the spatial localization of the outcome[16]. Pupil dilation and gaze direction were recorded at 500 Hz using an EyeLink 1000 Plus eye tracker, which was calibrated at the beginning of each run using a five-point calibration screen. The pupil dilation and gaze direction were extracted using the same method as in Pool et al.[16]. Briefly, the pupil data were preprocessed to remove eye blinks and extreme variations, and baseline corrected with a pre-stimulus baseline pupil size average of 1 s calculated for each trial and subtracted from each subsequent data point. The statistical analysis was conducted using the average pupil diameter between 0.5 and 1.8 s after stimulus onset. The averaged pupil diameter was adjusted to account for linear trends, independently of the trial type and changes related to switching responses from one side of the screen to the other. The dwell time in the ROIs was extracted

through the EyeMMV toolbox[68]. The ROIs were defined as squares centered on the food outcome delivery video, but 25% bigger than the actual video. Moreover, the index reflecting pupil dilation was adjusted by regressing out the gaze position on the screen and the index reflecting gaze direction was adjusted by regressing out the pupil size.

### Experimental design

The experimental procedure involved four main parts. First, participants selected their favorite sweet snack and salty snack. Second, they completed a Pavlovian conditioning task. Third, they underwent an outcome-devaluation procedure. Finally, they performed the test session under extinction. This procedure was the same as procedure used in Experiment 2 of Pool et al.[16], except that the Pavlovian task and the test session were administered in an fMRI scanner.

**Snack selection.** There were 16 snacks divided into two categories: sweet (M&M's, Buncha Crunch candy, almonds covered in cacao, Skittles, cereal covered in chocolate, raisins, yogurt-covered raisins, Milk Chocolate Morsels) and salty (roasted cashews, roasted peanuts, Goldfish, Simply Balanced Popcorn, cheese-flavored crackers, Ritz Bits cheese crackers, potato sticks, pretzel sticks). Participants tasted each sample and indicated their favorite salty snack and sweet snack (see Sup. Fig. 1 for pleasantness ratings). The participants' favorite snacks were used as outcomes during the Pavlovian conditioning task.

**Pavlovian conditioning session.** The task consisted of two learning runs lasting approximately 15 min each, both administered inside the scanner. Each run was composed of 60 trials for a total of 120 trials. At the beginning of each trial, four squares (6° visual angle each) highlighted by a white frame were displayed at the top and bottom horizontal center (18° visual angle on the x axis from the center) and the left and right vertical center (9° visual angle on the y axis from the center). These squares stayed on the screen for the whole duration of the trial.

Each trial was composed of (a) a cue presented for 1.5 s to 4.5 s in either the upper or lower white frames; (b) an empty screen with only the background white frames presented for 3 s; and (c) a video of the experimenter's hand delivering their favorite snack into a small bag lasting 3 s. When the video appeared in either the left or the right white frame (see Fig. 1), a picture depicting the small bag without any action was displayed on the opposite side of the screen. If no video was displayed, both sides displayed a picture of the small bag without any action. The inter-trial interval consisted of a fixation cross and was presented for 4 s to 8 s (uniformly distributed).

Participants were instructed to focus on the cue and to try to predict what was going to happen next. They were instructed to move their eyes freely around the computer screen, but to focus their gaze on the fixation cross during the inter-trial interval. Participants were asked to press the left key when the food outcome appeared on the left side of the screen and the right key when the food outcome appeared on the right side of the screen as quickly and accurately as possible. They were informed that the key-pressing task was a measure of their sustained attention, independent of the cue-outcome contingencies.

Contingencies were created so that one cue was more often associated with the delivery of the sweet food outcome on the left side of the screen (CS+ left sweet); one cue was more often associated with the delivery of the sweet food outcome on the right side of the screen (CS+ right sweet); one cue was more often associated with the delivery of the salty food outcome on the left side of the screen (CS+ left salty); one cue was more often associated with the delivery of the salty food outcome on the right side of the screen (CS+ right salty); and another cue was more often associated with no outcome delivery (CS-). Specifically, one cue predicted the delivery of a specific outcome 70% of the time (e.g., salty food outcome on the left), the remaining 30% of the

time the cue was followed by one of the other possible outcomes (e.g., 10% salty food outcome on the right; 10% sweet food outcome on the left; 10% no outcome; see Sup. Table 1). This created unexpected events where the identity of the outcome was as expected but the side was not (e.g., salty food outcome on the right), and unexpected events where the side of the outcome was as expected but the identity was not (e.g., sweet food outcome on the left).

Trials were presented in a pseudo-randomized order within participants with a maximum of three consecutive repetitions of the same kind of trial and with the first ten trials of the first session to be reinforced with the outcome they predicted more frequently (e.g., salty food to the left for CS+ left salty). The assignment of the neutral images to particular Pavlovian cue conditions (e.g., CS+ left salty, CS-) was counterbalanced across participants.

At the end of each run, there was a break taken outside the scanner during which participants received snacks to consume, which they collected during the task. The amount of food to consume depended on the amount of calories per individual piece of the snack the participant selected. Each participant ate the entirety of the snacks they received during the learning session.

**Outcome devaluation.** Participants were presented with a large bowl containing a very large amount of one of the two food outcomes used in the Pavlovian conditioning task. They were asked to eat it until they found the target food no longer palatable. We measured the level of hunger and food pleasantness with a visual analog scale before and after the selective satiation procedure. The food chosen for the devaluation procedure was counterbalanced across participants.

**Test session.** The test session was administered inside the scanner and was composed of a single run of 60 trials identical to the Pavlovian conditioning session, except that the outcome delivery was no longer visible. Participants were explicitly told that they would not be able to see any food outcome delivery video during this phase because the area where they were usually displayed would be hidden by two black patches for the whole duration of the session, but that they should assume that all the outcome deliveries would still occur as they had during the previous runs. They were also asked to press a key to guess under which of the two black patches the outcome delivery video was being displayed. The reason for using this strategy during the test session was to measure the influence of the outcome devaluation on the Pavlovian responses without confounding effects of the outcome itself, and at the same time to prevent the effects of behavioral extinction (e.g., disappearance of the conditioned responses due to lack of reinforcement) from happening too quickly[4,16].

### Statistical analysis
**Behavioral data**. Statistical analyses of the behavioral data were performed with R (version 4.0; R Core Team, 2019)[69]. We used the lmer4 package[70] and the lmerTest package[71] with the 'bobyqa' optimizer and set the number of model iterations to 1'000'000 to build the general linear mixed-effects models described hereafter.

To test for Pavlovian learning effects on the pupil dilation and the gaze direction, we built three contrasts. The first compared the dwell time in the left ROI after the perception of the CSs+ left (weight contrast +1) to the CSs+ right (weight contrast −0.5) and the CS- (weight contrast −0.5). The second compared the dwell time in the right ROI after the perception of the CSs+ right (weight contrast +1) to the CSs+ left (weight contrast −0.5) and the CS- (weight contrast −0.5). The third compared the pupil dilation during the perception of the CSs+ right (weight contrast +0.5) and the CSs+ left (weight contrast +0.5) and the CS- (weight contrast −1). For each of the dependent variables in the statistical model, we entered (1) the relevant contrast in interaction with (2) the run (first or second) as within-participants fixed factors. As random effects, we modeled random intercepts for participants (ID) and by-

participant random slopes for the relevant contrast in interaction with the run. The final models were built as follows (in lme4 syntax):

$$DV \sim contrast*run + (1 + contrast*run|ID) \qquad (1)$$

To test for Pavlovian learning effects on the reaction times (RT) to detect the location of the outcome delivery, we subset the databases to remove all responses to the CS- and then built two models. In the first one, we entered (1) the frequency of the outcome side for the associated CS (frequent or rare side) and (2) the run (first or second) as within fixed factors. In the second model, we entered (1) the frequency of the outcome identity for the associated CS (frequent or rare identity) and (2) the run (first or second) as within fixed factors. In both models, we entered random intercepts for participants (ID) and by-participant random slopes for the interaction between the two within-participants fixed factors. The final models were built as follows (in lme4 syntax):

$$RT \sim frequency*run + (1 + frequency*run|ID) \qquad (2)$$

To test for devaluation effects, we entered (1) the CS (valued or devalued) and (2) the session (pre- or post-devaluation) as within-participants fixed factors. We entered random intercepts for participants and by-participant random slopes for the main effects. The by-participant random slope for the interaction was not included in the random-effects structure as its inclusion led to model singularity, indicating overfitting[72]. We ran this model on two dependent variables (DVs): pupil dilation and the dwell time in the congruent ROI. For the pupil dilation, we averaged the pupil dilation during the CSs associated with valued outcome and the CSs associated with the devalued outcome and corrected it by subtracting the average pupil dilation during the CS-. For the gaze direction, we averaged dwell time allocated to the congruent ROI for all the CSs+ (dwell time in the right ROI after CS+ right and dwell time in the left ROI after CS+ left) for the CSs associated with the devalued outcome (CS devalued) and the CSs associated with the valued outcome (CS valued) and corrected it by subtracting the averaged dwell time during the CS- over both ROIs. We also ran this model on the pleasantness ratings of the food outcome (except that for the food outcome we entered food outcome [valued or devalued] rather than CS as a fixed effect). The final models were built as follows (in lme4 syntax):

$$DV \sim CS*session + (1 + CS + session|ID) \qquad (3)$$

We extracted Bayes factors through a linear mixed-effects Bayesian analysis using the BRMS package[73]. The models were estimated using Markov chain Monte Carlo (MCMC) sampling with 4 chains of 5000 iterations and a warm-up of 40'000 iterations. The dependent variables were scaled before being entered in the model. Priors parameters were set as Cauchy (*media* = 0.00, *scale* = 0.50) distributions. For one sample *t* tests, we computed Bayes factors with the function ttestBF from the BayesFactor package[74]. The Bayes factors reported for the main effects compared the model with the main effect in question versus the null model, while Bayes factors reported for the interaction effects compared the model including the interaction term to the model including all the other effects but the interaction term. Evidence in favor of the model of interest was considered anecdotal ($1 < BF_{10} < 3$), substantial ($3 < BF_{10} < 10$), strong ($10 < BF_{10} < 30$), very strong ($30 < BF_{10} < 100$) or decisive ($BF_{10} > 100$). Similarly, evidence in favor of the null model could also be qualified as anecdotal ($0.33 < BF_{10} < 1$), substantial ($0.1 < BF_{10} < 0.33$), strong ($0.033 < BF_{10} < 0.1$), very strong ($0.01 < BF_{10} < 0.033$) or decisive ($BF_{10} < 0.01$).

## Computational model analysis

We implemented two learning models. The first model learned value through reward prediction errors[18], while the second learned expectations about the perceptual properties of the outcome through state prediction errors[5,30].

The first model was a simple adaptation of the Rescorla-Wagner algorithm[18], where the reward prediction error was computed as follows:

$$\delta^{RPE} = R - V^{RW}_{(CS)} \tag{4}$$

The reward prediction error was used to update the expected value $V^{RW}$ of a given CS by multiplying it by a free parameter ($\alpha$; also referred to as learning rate) and summing this term with the CS current expected value, as follows:

$$V^{RW}_{(CS)} \leftarrow V^{RW}_{(CS)} + \alpha \times (\delta^{RPE}) \tag{5}$$

$R$ was coded as 1 when a food outcome was delivered – independently of the side (left or right) or the identity (sweet or salty) of the outcome. As participants were expecting to receive a food outcome at the start of the conditioning session (because of the instructions), we set the initial expected value $V_0$ for each of the five CSs to 0.5. The free parameter $\alpha$ was bounded within the range [0, 1].

The second model was an adaptation of the Forward model from Glascher et al.[30] and Schad et al.[5]. The model learns about the probability of transitioning from a given CS to another state such as the outcome with all its perceptual properties (unconditioned stimulus; US). These are called state-transition probabilities ($T_{(CS, US)}$), which are updated by state prediction errors ($\delta^{SPE}$). State prediction errors signal the discrepancy between ending up in a state and the expected probability of transitioning to that particular state. They were computed as follows:

$$\delta^{SPE} = 1 - T_{(CS,US)} \tag{6}$$

Because the five US outcome states (left sweet, right sweet, left salty, right salty, or nothing) were equally likely across conditions, we set the initial value of each transition probability ($T_0$) to 0.2.

State prediction error signals are not related to value, as they simply quantify how unexpected a particular state is given the previous state and are based on the perceptual properties (e.g., side and identity) of the outcome. Formally, the transition probabilities were updated as follows:

$$T_{(CS,US)} \leftarrow T_{(CS,US)} + \eta \times \delta^{SPE} \tag{7}$$

To keep the matrix normalized to total probabilities of one, transition probabilities for the other US states were updated as follows:

$$T_{(CS,US)} \leftarrow T_{(CS,US)} \times (1-\eta) \tag{8}$$

The free parameter $\eta$ is bounded within the range [0, 1] and acts as a learning rate determining the extent to which the state prediction error is weighted in the updating of the transition probabilities.

$$R = E[R|US] \tag{9}$$

This function attributed 1 to the rewarding outcomes (independently of their taste identity or location delivery) and 0 to the neutral outcome. The expected value was later computed for each of the CSs by multiplying the expected reward for each of the USs as follows:

$$V^{FW} = T_{(CS,US)} \times R \tag{10}$$

To determine the best-fitting learning rates $\alpha$ and $\eta$, we used maximum a posteriori estimation with the mfit toolbox (https://github.com/sjgershm/mfit)[75]. This consisted in finding the free parameter maximizing the likelihood of each participant's trial-by-trial pupil dilation data to the CSs given the model, constrained by a regularizing prior[75,76]. The free parameters were constrained with a $\beta(1.2, 1.2)$ prior distribution slightly favoring values that were in the middle of the parameter range. We used the trial-by-trial timeseries of predictive values to optimize the free parameters for the Rescorla-Wagner model ($V^{RW}$) and the adapted Forward model ($V^{FW}$; see Sup. Fig. 3).

## fMRI data acquisition

Acquisition was performed at the Caltech Brain Imaging Center (Pasadena, CA) using a 3-Tesla MRI system (Magnetom Tim Trio, Siemens Medical Solutions) using a 32-channel radio frequency coil. Functional images were acquired using a multi-band echo-planar imaging (EPI) sequence with the following parameters: 56 axial slices (whole-brain), A-P phase encoding, echo time (TE) = 30 ms, repetition time (TR) = 1000 ms, multi band acceleration of 4, field of view (FoV) = 200 × 200 mm, flip angle = 60°, 2.5 mm isotropic resolution, EPI factor of 80, echo spacing = 0.54 ms. Positive and negative polarity EPI-based field maps were collected before each run to allow geometric correction of the EPI data. Field maps were single band, TE = 50 ms, TR = 4800 ms, flip angle = 90°. We also acquired whole brain T1-weighted (T1$_w$) and T2-weighted (T2$_w$) anatomical images both with sagittal orientation (isotropic voxel size = 0.9 mm; FoV = 256 × 256 mm)

## fMRI data preprocessing

For the preprocessing, we used a combination of the Functional Magnetic Resonance Imaging of the Brain (FMRIB) Software Library (FSL, version 4.1)[77] and the Advanced Normalization Tools (ANTS, version 2.1)[78]. First, we reorientated and brain extracted all scans using fslreorient2std and the bet FSL commands, respectively. Following alignment of the T2 to the T1 (FSL flirt command), T1 and T2 scans were transformed into standard space using ANTs (CIT168 high resolution T1 and T2 templates[79]). Then, we used an fMRI independent component analysis (ICA) to remove artifacts. The multivariate exploratory linear optimized decomposition tool (MELODIC)[80] decomposes the raw BOLD signal into independent components (IC). These components were classified as signal or noise using a classifier that was trained on previous datasets from the laboratory. Noise components were removed from the signal using FSL's ICA-based X-noiseifier (FIX). We next applied field maps to correct geometric distortions. Field maps were extracted using FSL topup. De-noised functional scans were then unwarped with field maps using FSL fugue. Finally, we used ANTS to implement diffeomorphic co-registration of the preprocessed functional and structural images in the Montreal Neurological Institute (MNI) space, using the nearest-neighbor interpolation and leaving the functional images in their native resolution. Finally, we applied a spatial smoothing of 8 mm full-width half maximum (FWHM).

## fMRI data analysis

The Statistical Parametric Mapping software (SPM; version 12;[81]) was used to perform a random-effects univariate analysis on the voxels of the image time series following a two-stage approach to partition model residuals to take into account within- and between-participant variance[82,83].

For the first level, we specified a general linear model (GLM) for each participant. We used a high-pass filter cut-off of 1/128 Hz to eliminate possible low-frequency confounds. Each regressor of interest was derived from the onsets by constructing sets of stick functions (duration = 0) and were convoluted using a canonical hemodynamic

response function into the GLM to obtain weighted parameter estimates.

**Univariate analysis to define affective value and perceptual attributes ROIs.** The design matrix of the GLM contained the trials from the two learning runs. It consisted of the following regressors:

- the onsets of the CS and the anticipation screen parametrically modulated by the expected value
- the onsets of the outcome delivery parametrically modulated by (a) the presence/absence of the reward, (b) the reward prediction error, and (c) the state prediction error. All parametric modulators were allowed to compete for variance (no serial orthogonalization)
- the onsets of the left motor response
- the onsets of the right motor response

The resulting parameter estimates for the reward prediction errors and the state prediction errors were then entered into second-level one-sample $t$ tests to generate the random-effects level statistics.

The multiple comparisons correction was done using the Analysis of Functional magnetic resonance NeuroImages software (AFNI; version 20.2)[84]. We used the 3dFWHMx function to estimate the intrinsic spatial smoothness of each dimension. Then, we used the new 3dClustSim function[85] to create—via Monte Carlo simulation to form those estimates—a cluster extent threshold corrected for multiple comparisons at $p < 0.05$ for a height threshold of $p < 0.001$ within the whole brain. This provided an extended threshold of $k = 101$ for both the the reward prediction error and the state prediction error contrasts. The functional ROIs were defined using these corrected thresholds.

**Multivoxel pattern analysis to define identity and side ROIs.** First, we used SPM to build a GLM using the unsmoothed functional EPI volumes. We entered all the trials of the three runs into one design matrix. The design matrix included the following regressors:

- the onsets of the CS and the anticipation screen for each trial
- the onsets of the outcome delivery parametrically modulated by the presence/absence of the reward (in the first two runs only)
- the onsets of the left motor response
- the onsets of the right motor response

We extracted a T-map for each trial at the onsets of the CS and the anticipation screen and performed the classification analysis based on the parameter estimates of this GLM.

The classification analysis was performed in PyMVPA (version 2.5.0)[86]. We built two classifiers to decode two different attributes of the outcome associated with the CS during the CS presentation.

For the decoding of the side of the outcome, we first removed all the CS- trials and split the remaining trials into two sets. The first set included the two CSs sweet (i.e., the CSs predicting the delivery of the sweet outcome on the left and on the right). The second set was composed of the two CSs salty (i.e., the CSs predicting the delivery of the salty outcome on the right and on the left). We trained the classifier to decode the CS left versus right on one set and then tested the classification accuracy on the other set and vice-versa (see Fig. 5A, upper panel).

For the decoding of the outcome identity, we first removed all the CS- trials and split the remaining trials into two sets. The first set consisted of the two CSs left (i.e., the CSs predicting the delivery of the sweet outcome and the salty outcome on the left). The second set consisted of the two CSs right (i.e., the CSs predicting the delivery of the sweet outcome and the salty outcome on the right). We trained the classifier to decode the CS salty versus the CS sweet on one set and

then tested the classification accuracy on the other set and vice-versa (see Fig. 5 A, lower panel).

Classifier training and testing was done in a cross-validated manner with 2 folds and classification analyses were performed with a linear support vector machine (SVM) classifier.

This approach (see[27,87]) allowed us to target the voxels representing the identity or the side of the associated outcome independently of the visual features of the CSs themselves.

We performed a whole brain searchlight analysis with a spherical searchlight, using a radius of 3 voxels. The SVM cost/penalty parameter C was set to 1.0 for all searchlight analyses. The classification accuracy of each searchlight was assigned to the center voxel of the sphere. Before second-level analyses, individual accuracy maps were smoothed with a Gaussian smoothing kernel of 8 mm (FWHM). To test the global null hypothesis that there is no information in the test population, we used a one-sample $t$ test testing whether the classifier performance was above 50% (i.e., chance level). None of the second level contrasts survived whole brain correction. Therefore, we defined the functional ROIs involved in the representation of the side and the identity of the outcome by using a lenient threshold of $p_{uncorrected} = 0.005$, $k = 100$. The map of the side decoding was masked to remove motor movements and residual eye movements. The motor movements mask was created based on the GLM to define affective value and perceptual attributes. More specifically by the contrast: *left motor response > right motor response* and the contrast: *right motor response > left motor response*. Two clusters from motor areas (left and right) were extracted with a threshold of $p > 0.005$ and used as a mask. The residual eye movements mask was created based on the GLM to define identity and side. We used the second level t-map to decode side. Two clusters over from the residual activations from the eyeball (left and right) were extracted with a threshold of $p < 0.01$ and used as a mask. The thresholds of the masks were set to be one step more lenient than their respective analysis of interest (i.e., $p < 0.005$ for the univariate analysis at $p < 0.001$ and $p < 0.01$ for the multivariate analysis at $p < 0.005$).

**Univariate analysis testing devaluation effects.** We built two different GLMs to test for devaluation effects at (1) the time of the outcome delivery in the ROIs defined based on the state and reward prediction errors, and (2) at the time of the CS onset in the ROIs defined based on the outcome identity and outcome side decoding.

The first GLM was built using SPM on the test run only and consisted of the following regressors:

- the onsets of the CS and the anticipation screen
- the onsets of the delivery of the valued outcome predicted by the CSs+ hidden by the black patches
- the onsets of the delivery of the devalued outcome predicted by the CSs+ hidden by the black patches
- the onsets of the no-outcome delivery predicted by the CS- hidden by the black patches (half of the trials sampled randomly)
- the onsets of the no-outcome delivery predicted by the CS- hidden by the black patches (the other half of the trials)
- the onsets of the left motor response
- the onsets of the right motor response

From the ROIs, we extracted the $\beta$ estimates of the contrast between the valued outcome versus the no outcome (i.e., *valued outcome > no outcome*) in the first subset of trials and the devalued outcome versus the no outcome (i.e., *devalued outcome > no outcome*) in the second subset of trials. We thereby obtained the $\beta$ estimates for (1) the valued outcome and (2) the devalued outcome.

The $\beta$ estimates extracted from the relevant ROIs were entered in R and analyzed with a repeated-measures regression using the package nlme with the lme function[88]. We entered outcome value

(valued or devalued) as a fixed factor and participant (ID) as a random-effect factor. The model was built as follows (in the nlme syntax):

$$\beta \, estimates \sim value, random = \sim value | ID \qquad (11)$$

The p-value threshold was also adjusted for the number of tests. Specifically, the level of statistical significance $\alpha$ was divided by the number of ROIs for a given contrast in which the devaluation effect was tested.

The second GLM was built using SPM by entering the last learning run and the test run in a design matrix. The GLM included the following regressors:

- the onsets of the CSs+ and the anticipation screen associated with the delivery of the valued outcome
- the onsets of the CSs+ and the anticipation screen associated with the delivery of the devalued outcome
- the onsets of the CS- and the anticipation screen (half of the trials sampled randomly)
- the onsets of the CS- and the anticipation screen (the other half of the trials)
- the onsets of the outcome delivery modulated by the presence/absence of the reward (for the learning run only)
- the onsets of the left motor response
- the onsets of the right motor response

From the ROIs, we extracted the beta estimates for: (1) the CS+ valued pre-devaluation compared to the CS- pre-devaluation (i.e., *CS+ valued pre > CS- pre*) in the first subset of trials; (2) The CS+ valued post-devaluation compared to the CS- post-devaluation (i.e., *CS+ valued post > CS- post*) in the first subset of trials; (3) The CS+ devalued pre-devaluation compared to the CS- pre-devaluation (i.e., *CS+ devalued pre > CS- pre*) in the second subset of trials; (4) The CS+ devalued post-devaluation compared to the CS- post-devaluation (i.e., *CS+ devalued post > CS- post*) in the second subset of trials.

Via this procedure, we obtained a $\beta$ estimate for (1) the CS+ valued pre-devaluation; (2) the CS+ valued post-devaluation; (3) the CS+ devalued pre-devaluation; and (4) the CS+ devalued post-devaluation.

The $\beta$ estimates extracted from the relevant ROIs were entered in R. We computed the difference in the $\beta$ estimates pre- and post-devaluation for each of the CSs. We then analyzed the differential $\beta$ estimates in a repeated-measures regression using the package nlme with the lme function[88]. We entered CS value (valued or devalued) as a fixed factor and participant (ID) as a random-effect factor. The model was built as follows (in the nlme syntax):

$$\Delta \beta \, estimates \sim value, random = \sim value | ID \qquad (12)$$

The *p*-value threshold was also adjusted for the number of tests by dividing the level of statistical significance $\alpha$ by the number of ROIs for a given contrast for which the devaluation effect was tested.

**Brain-behavior correlations.** To test whether reward prediction errors and state prediction errors are associated with predictions about the side and the identity attributes of the outcome, we correlated the individual $\beta$ estimates from the state and reward prediction errors with the individual difference in reaction times to detect the outcome. More precisely, we computed two indices based on reactions times.

The first index reflects the unexpected side effect. To compute this index, we removed the CS- trials and used only trials where the identity was the one most often predicted by the CS but the side of the outcome delivery varied. We subtracted the average reaction time in trials where the side of the outcome was the same as the one most often predicted by the CS from the average reaction time in trials where the side was different from the one most often predicted by the CS.

The second index reflects the unexpected identity effect. This index was calculated by removing the CS- trials and using only the trials where the side was the one most often predicted by the CS but the outcome identity varied. We subtracted the average reaction time in trials where the identity was the same as the one most often predicted by the CS from the average reaction time in trials where the identity was different from the one most often predicted by the CS.

### Reporting summary
Further information on research design is available in the Nature Portfolio Reporting Summary linked to this article.

## Data availability
The fMRI data generated in this study has been deposited in the YARETA database[89] under accession code: https://doi.org/10.26037/yareta:dyhmmxkwkfbwvaq4yotxziszva. Source data are provided with this paper.

## Code availability
The code used to generate the figures and the results reported in this manuscript is available on the following repository: https://github.com/evapool/PavlovianPredictions/; Zenodo https://doi.org/10.5281/zenodo.10004873.

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

## Acknowledgements

We would like to thank Dr. Yoann Stussi and Dr. Vanessa Sennwald for helpful discussions. This work was supported by an Ambizione research grant (project PZPGP1 193120) to ERP and a National Institute on Drug Abuse (R01DA040011) grant to JPOD.

## Author contributions

E.R.P., J.P.O and W.M.P. conceptualized the work and the experimental design. E.R.P. and W.M.P. collected the data. E.R.P. and L.C. performed the analysis. E.R.P. and J.P.O wrote the original draft of the manuscript. All authors contributed to reviewing and editing of the manuscript.

## Competing interests

The authors declare no competing interests.
