## [Peer Review File · Nature Communications]

Neural substrates of parallel devaluation-sensitive and devaluation-insensitive Pavlovian learning in humansReviewer #1 (Remarks to the Author):

This manuscript by Pool et al tests for devaluation-sensitive and devaluation-insensitive brain responses during Pavlovian learning. The mapping between model-based versus model free reinforcement learning and goal-directed versus habitual instrumental behavior is well established. In contrast, it is unclear whether a similar mapping exists in Pavlovian learning. This manuscript fills this gap by testing which neural responses and representations are sensitive to devaluation, the gold-standard for distinguishing between habitual and goal-directed behavior.

The authors use a clever behavioral task with two different reward types to examine neural correlates of learning signals and predictive representations that may support model-free and model-based Pavlovian responses. To test whether these neural signals are model-based or model free, the authors devalue one of the rewards and show that a subset of these signals are devaluation sensitive. Importantly, this subset does not fully line up with how we traditionally think about these different signals. For instance, they show that reward prediction error (PE) signaling in the ventral striatum is devaluation sensitive, questioning the traditional classification of this signal as model-free.

This is an important contribution. The study is well designed, and the analyses are clear and straightforward. The manuscript is well written. I do have one conceptual point about the framing of the introduction, as well as some specific questions that I hope the authors can address.

Major points:

1. I find the framing of the study in the introduction somewhat problematic. The clear dichotomy between model-based and model-free works well for behavior, and using devaluation to tease them apart is perfectly fine. But I am not sure this can be directly translated to neural responses. It certainly works well for reward PEs: Devaluation insensitive reward PEs means they are computed based on cached values, whereas devaluation sensitive reward PE indicates that they are computed using model-based/inferred value estimates (e.g., Sadacca et al 2016, eLife). In contrast, it does not work so well for predicted reward identity (or side) or state PEs. There are many instances where you wouldn't expect a model-based signal to be sensitive to devaluation. For example, assuming state PEs are a model-based signal that supports learning of a model that may be used for model-based inference, you would not expect this signal to be devaluation sensitive. You want this signal to construct a model of the world as it is and not only based on your current state. Similarly, not all predictive representations of reward identity should be devaluation sensitive because we need an invariant model of the world to do model-based inference (we should not update the model that stimulus A predicts chocolate just because we ate a bunch of chocolate). So ideally, both of these model-based signals should not be devaluation sensitive. On the other hand, we expect other model-based signals to be devaluation sensitive. For instance, devaluation sensitive representations of predicted reward identity would suggest that these signals integrate the identity and the inferred value of the expected outcome – the output of model-based inference. My point is that there is no clear a priori reason for specific model-based representations in the brain to be devaluation sensitive or not. However, knowing which signals are sensitive and which are not is important and helps us to better understand the role of these brain areas in behavior. The introduction covers a lot of ground that is only partially relevant to the current question. I think the paper would be improved if it focused more on the key questions and made explained what different results would mean for our understanding of these brain areas in Pavlovian learning.

2. Figure 3: I think most readers would appreciate an explanation of the rationale for testing BOLD responses to omitted outcomes in extinction to determine neural responses to reward and state PE. How can we interpret these results? Also, only the response difference (valued (CS+ - CS-) minus devalued (CS+ - CS-)) are shown, but it would be interesting to see the responses (CS+ minus CS-) individually for devalued and devalued omitted outcomes.

3. The current study examines two different state-based (value-neutral) signals: identity and side. It is very possible that they are not devaluation sensitive to the same degree. In fact, only reward identity is affected by the current devaluation procedure, and so you could expect differences in how signals for side and identity are modulated by devaluation. The decoding analysis based on CS-evoked fMRI responses distinguishes between identity and side, but I did not see a similar analysis for state PEs. Examining differences between identity and location PE, and how they

change with devaluation would be very interesting.

4. The results section (page 13) would benefit from a description of “the unexpected side effects and the unexpected identity effects as measured with reaction times” before using them for the correlation analysis with the state-PE responses. Something like the last section of the methods section would be helpful. Also, from which areas were the state-PE betas (shown in Figure 4) extracted? Please describe this in the main text and clarify that these analyses focus on data from the acquisition phase.

5. I am not sure I fully understand the logic of comparing pre-post differences in CS-evoked univariate responses between devalued and non-devalued rewards in areas that show significant decoding for side and reward identity. I’d have expected to see a comparison of decoding accuracy for reward identity or side between pre- and post-devaluation, though I can imagine that this is challenging given the limited data.

6. What is the distribution of the fitted learning rates from the two models? Are they correlated across subjects? Also, how well did the two models (with best fitting learning rates) explain the pupil data? Assuming identical learning rates, how different are the trial-by-trial estimates of V_{FW} and V_{RW} ? I also don’t fully understand equation (9). What values can ‘R’ take on the left and right hand side of the equation? Where do the values for $E[R/US]$ come from in this model? Is this an indicator function that produces 1 for rewarding US and 0 for neutral US or are these objective probabilities from the task design?

Minor comments

1. Please clarify in the main text that responses in Figure 3A and C were based on data from the acquisition phase of the task.

2. Was subjects’ food intake prior to the experiment controlled? How much did subjects eat (in calories) during the devaluation procedure?

3. Did pleasantness ratings differ between the two food identities? And if so, does this introduce a confound in the analysis?

4. There were 60 trials per conditioning run. How many trials for each of the five CSs and for each of the three deviant outcomes per CS? What were the deviant outcome for the neutral CS?

5. Figure 4: If I understand this plot correctly, correlations of state PEs with both behavioral measures (location and identity) are shown in panel A, whereas panel B shows correlations with reward PE. If this is correct, then please correct the main text on the bottom of page 13 to reflect this (this text currently suggests that correlations with state PE for identity and location effects are shown in panel A and B, respectively).

6. The authors are aware that the side decoding analysis is confounded by motor responses and eye movements. Please describe in the methods section how exactly the maps from the side decoding analysis were masked to remove signals related to motor responses and eye movements.

7. Training and testing the classifier on data from the same runs (even when training and testing on different trials) may introduce biases. I assume the authors used this approach to maximize the amount of training data. However, it would be reassuring if they could show that the basic results hold up when training and testing the classifier on trials from different runs. Perhaps this control analysis could be performed in the ROIs from the original analysis.

8. I was not clear what the authors meant by “concatenated” when describing the GLMs. First, I assumed that they appended data from all runs to estimate a single parameter estimate for a given condition for all runs. But this would not work for the analysis described on page 42, which separates responses pre- and post-devaluation. Do the authors mean that the GLM included data from multiple runs but each condition was still modeled with different regressors to obtain independent parameter estimates? If the latter is the case (which the default in SPM), I would suggest removing the term concatenation as it is typically used to refer to what is done using

spm_fmri_concatenate.

9. Please check the reference list for typos.

Reviewer #2 (Remarks to the Author):

In this manuscript authors aimed to differentiate brain areas involved in learning and encoding associations that are sensitive to changes in the value of an outcome from those that are not sensitive to such changes. They find that, regions whose activity correlates with reward prediction errors are sensitive to outcome devaluation, challenging the assumption that reward prediction errors are exclusively model-free. Similarly, brain areas correlating with state prediction errors were found to be less sensitive to outcome devaluation, challenging the assumption that state prediction errors are model-based.

Such findings appear noteworthy and of broad interest. The manuscript is clearly written, the methodology is sound and the conclusions are supported by the data. I only have a few minor suggestions that should be considered before publication.

- p. 9-10 "outcome devaluation induced changes": what about RTs, did outcome devaluation have any effect on RTs at Test?

- p. 28 "the video appeared on the left or right white frame", what appeared on the other frame? Fig. 1a seems to show that the background of the clip appears on the other frame, please add this to the description. Is this background also presented in "no outcome" trials (p. 29)? You should specify this.

- p. 28 please specify that participants needed to press a button corresponding to the side of outcome delivery, and that RTs were collected.

- p. 29 "At the end of each run, there was a break to be taken outside the scanner during which participants received a bag containing the snacks they collected during the task, to consume." Did this quantity vary between subjects and/or runs and/or outcomes? How much was the variations between subjects and/or runs and/or outcomes? Could variations in the amount of snacks eaten during the conditioning session, e.g. if more snacks were eaten for one outcome than the other, then have an effect on devaluation and the subsequent test session?

- p. 30 "They were also asked to press a key to indicate which one of the two black patches was obscuring the outcome delivery video."

Response to reviewers

Reviewer 1

This manuscript by Pool et al tests for devaluation-sensitive and devaluation-insensitive brain responses during Pavlovian learning. The mapping between model-based versus model free reinforcement learning and goal-directed versus habitual instrumental behavior is well established. In contrast, it is unclear whether a similar mapping exists in Pavlovian learning. This manuscript fills this gap by testing which neural responses and representations are sensitive to devaluation, the gold-standard for distinguishing between habitual and goal-directed behavior. The authors use a clever behavioral task with two different reward types to examine neural correlates of learning signals and predictive representations that may support model-free and model-based Pavlovian responses. To test whether these neural signals are model-based or model free, the authors devalue one of the rewards and show that a subset of these signals are devaluation sensitive. Importantly, this subset does not fully line up with how we traditionally think about these different signals. For instance, they show that reward prediction error (PE) signaling in the ventral striatum is devaluation sensitive, questioning the traditional classification of this signal as model-free. This is an important contribution. The study is well designed, and the analyses are clear and straightforward. The manuscript is well written. I do have one conceptual point about the framing of the introduction, as well as some specific questions that I hope the authors can address.

Authors' response: Thank you for your helpful review, we strongly appreciate your positive and relevant comments. We address each of them below.

Reviewer 1, comment 1: *I find the framing of the study in the introduction somewhat problematic. The clear dichotomy between model-based and model-free works well for behavior, and using devaluation to tease them*

apart is perfectly fine. But I am not sure this can be directly translated to neural responses. It certainly works well for reward PEs: Devaluation insensitive reward PEs means they are computed based on cached values, whereas devaluation sensitive reward PE indicates that they are computed using model-based/inferred value estimates (e.g., Sadacca et al 2016, eLife). In contrast, it does not work so well for predicted reward identity (or side) or state PEs. There are many instances where you wouldn't expect a model-based signal to be sensitive to devaluation. For example, assuming state PEs are a model-based signal that supports learning of a model that may be used for model-based inference, you would not expect this signal to be devaluation sensitive. You want this signal to construct a model of the world as it is and not only based on your current state. Similarly, not all predictive representations of reward identity should be devaluation sensitive because we need an invariant model of the world to do model-based inference (we should not update the model that stimulus A predicts chocolate just because we ate a bunch or chocolate). So ideally, both of these model-based signals should not be devaluation sensitive. On the other hand, we expect other model-based signals to be devaluation sensitive. For instance, devaluation sensitive representations of predicted reward identity would suggest that these signals integrate the identity and the inferred value of the expected outcome – the output of model-based inference. My point is that there is no clear a priori reason for specific model-based representations in the brain to be devaluation sensitive or not. However, knowing which signals are sensitive and which are not is important and helps us to better understand the role of these brain areas in behavior. The introduction covers a lot of ground that is only partially relevant to the current question. I think the paper would be improved if it focused more on the key questions and made explained what different results would mean for our understanding of these brain areas in Pavlovian learning

Authors' response: Thank you very much for raising this important point. We fully agree that our framing of model-based predictions would benefit from being further clarified. Indeed, we agree that there likely exists multiple model-based signals with different sensitivities to the outcome value and that identifying which signals are involved in stable value-invariant representations and signals that integrate changes in outcome value is important to elucidate the brain mechanisms involved in inflexible Pavlovian responses. We fully re-wrote the second part of the introduction, we removed the unnecessary element and integrated your suggestions. The relevant part of the revised introduction now reads as follows (p. 4):

“Devaluation insensitive behaviors are often suggested to rely on brain signals that can be approximated through model-free reinforcement learning algorithms, which use reward prediction errors to make predictions based on cached values [14, 46]. A key empirical test of this hypothesis as applied to Pavlovian conditioning would be whether brain regions correlating with model-free reinforcement learning based on reward prediction errors are sensitive to changes in outcome value. If brain regions sensitive to reward prediction errors are found to be insensitive to devaluation, this could provide evidence for the role of model-free reinforcement-learning in the acquisition of devaluation insensitive Pavlovian behaviors in humans. On the other hand, if such reward prediction error signal coding brain regions turn out to actually be sensitive to outcome devaluation, then this could provide evidence to suggest that reward prediction error-based learning is not model-free during Pavlovian conditioning.

Within a model-based framework, some computations would be expected to be devaluation sensitive while others would not. Model-based predictive representations of outcome-value would be devaluation sensitive by definition, as these representations are hypothesized to emerge by integrating knowledge of stimulus-stimulus associations with knowledge about current expected outcome value. On the other hand, internal representations of the cognitive model itself, should not be sensitive to changes in outcome-value, such as for instance, information about where in the environment an outcome is expected to occur.”

We also added the relevant reference to Sadacca et al., (2016) in the discussion section.

Reviewer 1, comment 2, part A: *Figure 3: I think most readers would appreciate an explanation of the rationale for testing BOLD responses to omitted outcomes in extinction to determine neural responses to reward and state PE. How can we interpret these results?*

Authors’ response: We have now elaborated on the rationale for testing BOLD responses to omitted outcomes in extinction. The relevant text now reads as follows (p. 11):

"To test for devaluation effects inside these ROIs, we compared activity while participants expected a valued versus a devalued outcome, during the run after the devaluation procedure. We used a pseudo-extinction procedure, whereby the visual presentation of the outcomes was obscured behind two black patch covers present at the time of the outcome delivery. The use of pseudo-extinction is a crucial manipulation that prevents rapid relearning of a CS's expected value via the newly devalued outcome. Thus, this procedure allows predictive representations that are linked to the incentive value of the predicted outcome to be dissociated from those associated with outcome-value insensitive representations."

Reviewer 1, comment 2, part B: *Also, only the response difference (valued (CS+ - CS-) minus devalued (CS+ - CS-)) are shown, but it would be interesting to see the responses (CS+ minus CS-) individually for devalued and devalued omitted outcomes.*

Authors' response: In the supplementary material we added the requested figure (see Sup. Fig. 6, p. 7 supplementary material).

For the sake of completeness, in the supplementary material, we also added figures displaying the valued and devalued conditions separately for each devaluation test (e.g., pupil, gaze direction, state and reward prediction errors ROI, side and identity ROI).

Reviewer 1, comment 3: *The current study examines two different state-based (value-neutral) signals: identity and side. It is very possible that they are not devaluation sensitive to the same degree. In fact, only reward identity is affected by the current devaluation procedure, and so you could expect differences in how signals for side and identity are modulated by devaluation. The decoding analysis based on CS-evoked fMRI responses distinguishes between identity and side, but I did not see a similar analysis for state PEs. Examining differences between identity and location PE, and how they change with devaluation would be very interesting.*

Authors' response: Thank you for this suggestion, we fully agree that it would indeed very be elegant to have a parallel between the mpva approach

Sup. Fig. 6 : Sensitivity to outcome devaluation in reward prediction errors ROI. Betas for the valued and the devalued contrast in the midbrain ROI the ventral striatum / sgACC ROI (VS), the ventromedial prefrontal cortex ROI (vmPFC). The "valued contrast" was defined as the difference in the BOLD signal during the outcome delivery (displayed behind two black patches) after the perception of the valued CS+ versus the CS-. The "devalued contrast" was defined as the difference in the BOLD signal during the outcome delivery (displayed behind two black patches) after the perception of the devalued CS+ versus the CS-. Error bars represent 95% confidence interval. Double asterisks indicate statistically significant differences between conditions that survive correction for multiple comparisons.

decoding identity and side, and the analysis of the prediction errors. We tried to follow up on this suggestions with multiple analyses.

First, we built two different models: one that tracked transition probabilities of sweet and salty outcomes without distinguishing between left and right, and one that tracked the transition probabilities of left and right outcomes without distinguishing between sweet and salty. This way, we were able to derive a state identity prediction error regressor (idSPE) and a state side prediction error regressor (sideSPE). However, these two regressors were highly correlated (i.e., up to $r = 0.769$), because they had similar predictions due to the CS- condition (e.g., high values when the CS- was followed by an outcome and when one of the CS+ was followed by no outcome). Indeed, in each run there were only 4 deviant outcomes specific to the side or identity and 7 that were common because they involved the the CS- condition or the absence of the reward as an outcome (see Table 1 in our answer to your minor comment 4).

Second, to de-correlate these signals, we created two regressors: the first regressor signalled a deviance specific to the identity (i.e., 1 if it was deviant and 0 if it was not) and the second one signalled a deviance specific

to the side. Since this analysis relies on 4 items per run on two runs only, for reasons of limited statistical power, we only tested these regressors in the voxels that showed significant activation for the global SPE contrast. We found significant activations in voxels of the VTA for the identity deviance and significant activations in the in MFG, bilateral OFC and VTA for the side deviance. To test whether these activations indeed reflected side and identity prediction errors, we correlated them with the reaction times reflecting the unexpected taste identity effect and the unexpected side effect. Unfortunately, we did not find evidence for any brain-behavior correlations. For the sake of completeness, we also tested for devaluation effects in these voxels. We did not find statistical evidence for sensitivity to devaluation for the identity SPE nor for the side SPE (see Fig. for Revisions 1).

We did not include these analyses in the manuscript because of the limitations of the present experimental design for isolating identity and state SPE due to the very limited number of trials of each type. Nonetheless, we added a paragraph in the discussion explaining how this aspect is a limitation of the current study that could be addressed by future experimental work. The relevant paragraph reads as follow (p. 24):

"A complementary approach to investigate the representation of the outcome side and the outcome identity would be to derive specific state prediction errors from models that only track the taste identity of the outcome (irrespective of the side of the delivery) and models that only track the side of the outcome delivery (irrespective of the taste identity of the outcome). In the present study, due to a limited amount trials available to test for these specific conditions in our design, we could not use this approach; but future studies could try to design experiments allowing to parse different aspects of the outcome based on different kinds of state prediction errors. "

Reviewer 1, comment 4: *The results section (page 13) would benefit from a description of “the unexpected side effects and the unexpected identity effects as measured with reaction times” before using them for the correlation analysis with the state-PE responses. Something like the last section of the methods section would be helpful. Also, from which areas were the state-PE betas (shown in Figure 4) extracted? Please describe this in the main text and clarify that these analyses focus on data from the acquisition phase.*

Fig. for Revisions 1 : Identity and Side Prediction Errors. **A** Correlation between the magnitude of the side state prediction error (i.e., side SPE) effect and difference in the reaction times to detect the outcome when the side was unexpected vs expected. **B** Correlation between the magnitude of the identity state prediction error (i.e., identity SPE) effect and difference in the reaction times to detect the outcome when the taste identity was unexpected vs expected. **C** Sensitivity to outcome devaluation (betas for the devalued contrast - betas for valued contrast) in the ROIs based on the side SPE (in yellow) and the identity SPE (in red). The ROI covered parts of the superior frontal gyrus (SFG), of the bilateral orbitofrontal/anterior insula (OFC), and of the midbrain (VTA). The "valued contrast" was defined as the difference in the BOLD signal during the outcome delivery (displayed behind two black patches) after the perception of the valued CS+ versus the CS-. The "devalued contrast" was defined as the difference in the BOLD signal during the outcome delivery (displayed behind two black patches) after the perception of the devalued CS+ versus the CS-. Error bars represent 95% confidence interval.

Authors' response: We have modified the text of the result section by adding the requested information. The relevant text now reads as follows (pp. 13-15) :

To test to what extent these two aspects are reflected in the state prediction error brain signals, we extracted the effect of the state prediction error from the state prediction error ROIs and averaged it across the different ROIs. Then we correlated the averaged effect of the state prediction error against the *unexpected side effect* and the *unexpected identity effect* as measured with the reaction times during the two runs of the acquisition phase. More precisely, to compute a reaction time index reflecting the *unexpected side effect*, we only used the trials where the identity was the one most often predicted by the CS but the side of the outcome varied. We subtracted the average reaction time in trials where the side of the outcome was the same as that most often predicted by the CS from the average reaction time in trials where the side was different from that most often predicted by the CS. To compute the reaction time index reflecting the *unexpected identity effect*, we only used the trials where the side was the one most often predicted by the CS but the identity of the outcome varied. We subtracted the average reaction time in trials where the identity was the same as that most often predicted by the CS from the average reaction time in trials where the identity was different from that most often predicted by the CS.

We also modified the figure legend by adding the same relevant information. The figure legend now reads as follows (p. 16):

Correlation between state and reward prediction error and unexpected effects measured with reaction times during the acquisition phase. **A)** Correlation between the magnitude of the state prediction error (i.e., SPE) effect and difference in the reaction times to detect the outcome when the side was unexpected vs expected (in yellow) and when the taste identity was unexpected vs expected (in red). **B)** Correlation between the magnitude of the reward prediction error (i.e., RPE) effect and difference in the reaction times (i.e., RT) to detect

the outcome when the side was unexpected vs expected (in yellow) and when the taste identity was unexpected vs expected (in red).

Reviewer 1, comment 5: *I am not sure I fully understand the logic of comparing pre-post differences in CS-evoked univariate responses between devalued and non-devalued rewards in areas that show significant decoding for side and reward identity. I'd have expected to see a comparison of decoding accuracy for reward identity or side between pre- and post-devaluation, though I can imagine that this is challenging given the limited data.*

Authors' response: Indeed, previous studies investigating devaluation effects in neural representations underpinning goal-directed behavior (see Howard et al., 2015; J. Neurosci.) used the decoding pre-post decoding approach you suggested. You are also correct in pointing out that the data of our experimental task are not appropriate this analytical approach, as we have a limited amount of data, in particular in the post devaluation run. The logic for using the decoding approach to identify the regions of interest is the same as the one we used for the state and reward prediction errors (see our answer to your comment 2). We used the decoding approach to identify the regions involved in learning the association between the CS and different aspects of the outcome. This allowed us to run the univariate devaluation test in select ROI which were informed by the decoding approach. Of course, one limitation of this approach is that the univariate test of devaluation might miss changes that are coded by multivoxel pattern activations. However, we did find evidence for devaluation effects in ROIs defined based on outcome identity decoding, which suggests that the univariate approach might be a sufficiently sensitive test. However, we also want to underline that the findings based on the decoding are a supplementary analysis which provides new preliminary data congruent with the existing literature (Howard et al., 2015; J. Neurosci.) but also needs to be confirmed in future studies. We added a paragraph in the discussion mentioning this limitation. the relevant text now reads as follows (pp. 23-24)

"Please note that this interpretation relies on a supplementary analysis carried out with a lenient threshold to define the regions of interest, therefore further empirical confirmation will be

prudent before drawing more definitive conclusions. It is also important to underline that our devaluation sensitivity test relied on a univariate approach which would not be able to detect changes in outcome value representations manifesting via changes in multi-voxel patterns."

Reviewer 1, comment 6: *What is the distribution of the fitted learning rates from the two models? Are they correlated across subjects? Also, how well did the two models (with best fitting learning rates) explain the pupil data? Assuming identical learning rates, how different are the trial-by trial estimates of V^{FW} and V^{RW} ? I also don't fully understand equation (9). What values can 'R' take on the left and right hand side of the equation? Where do the values for $E[R/US]$ come from in this model? Is this an indicator function that produces 1 for rewarding US and 0 for neutral US or are these objective probabilities from the task design?*

Authors' response: The fitted learning rates from the two models were not significantly correlated across subject. To provide information about the distribution of the fitted learning rate, we created a new figure (see Sup. Fig. 3) displaying the distribution the individual estimates of the learning rates from the two models. We added this figure to the supplementary materials (p. 6 supplementary material):

Sup. Fig. 3 : Distribution of the estimated value of the free parameters. Estimated value of the learning parameters for the Forward model (η) and Rescorla Wagner (α). Error bars represent 95% confidence interval.

The mean correlation across all participants between trial-by-trial regressor value derived from RW and the one derived from FW was of $r = 0.82$. This is the main reason why we did not use them as regressors for the analysis of the brain activity, but instead we used the trial-by-trial regressors of reward prediction error derived from the RW model and state prediction error derived from the FWD model which were not highly correlated (average $r = 0.166$).

To test how well the two models predicted the pupil data, we derive the trial-by-trial value regressors from the two models (with the best fitting parameters) and used them as predictors in a linear mixed regression. The analysis revealed that the value regressor derived from the FWD model accounted well the pupil data, while the value regressor derived from the RW model did not reach statistical significance (though trended at $p = 0.053$), however the BF provided evidence favoring the effect. We now report these analysis in the supplementary materials. The relevant section reads as follows (pp. 3-4 supplementary material):

"Pupil dilation and model-based value regressors

As a control analysis, we tested how well the two model-based regressors explain the pupil dilation responses to the CSs. We first derived the value regressor from the Rescorla-Wagner model and the Forward model using the best fitting learning rate for each one of the participants. We then entered the value based regressor as within-participants fixed factors. As random effects, we modeled random intercepts for participants (ID) and by-participant random slopes for the value regressor. The final models were built as follows (in lme4 syntax):

$$pupil \sim value + (1 + value | ID) \quad (1)$$

This analysis revealed that the value regressor derived from the Forward model significantly explain the pupil responses ($\beta = 0.062$, $SE = 0.024$, 95% CI = [0.0150, 0.109], $p = 0.015$, $BF_{10} = 4.05$) while the regressor derived from the Rescorla-Wagner did not reach statistical significance ($\beta = 0.087$, $SE = 0.042$, 95% CI = [0.003, 0.1710], $p = 0.053$, $BF_{10} = 2.52$)."

Concerning your question about the equation (9), you understood correctly that is a function that produces 1 for a rewarding US (independently

of their delivery location or their taste identity) and 0 for neutral. That does not reflect the objective probabilities of the task design. We now clarify this in the method section. The relevant text reads as follows (p. 36):

"This function attributed 1 to the rewarding outcomes (independently of their taste identity or location delivery) and 0 to the neutral outcome."

Reviewer 1, minor comment 1: *Please clarify in the main text that responses in Figure 3A and C were based on data from the acquisition phase of the task.*

Authors' response: Thank you for this helpful suggestion. We added the explanation in the revised manuscript. The relevant text reads as follows (p. 11):

"To identify brain ROIs separately involved in implementing Pavlovian predictions about the affective value and perceptual attributes of the outcome, respectively, we derived trial-by-trial prediction errors during the first two runs from two models: [...]"

Reviewer 1, minor comment 2: *Was subjects' food intake prior to the experiment controlled? How much did subjects eat (in calories) during the devaluation procedure?*

Authors' response: Food intake prior to the experiment was controlled; we asked participants to refrain from eating for 6 hours prior to the experimental session. We added this information in the revised manuscript. The relevant text reads as follows (p. 27):

" Participants were prescreened to ensure they were not dieting and they were asked not to eat for at least 6 h before the experimental session (they were allowed to drink water)"

During the experimental procedure we controlled the amount of calories consumed during each run as follows. Each run had exactly the same number of outcome videos displayed. However, the different snacks used as outcomes were selected based on the participant’s preferences and varied between participants in terms of calories. We accounted for this difference by adapting correspondence between the amount of food consumed at the end of each session to the amount of calories of each snack. this correspondence was proportional and varied from 1:2 to 1:6 according to the amount of calories per individual piece of the snack selected by the participant (see also our answer to comment 4 of reviewer 2). During the devaluation procedure, participants were free to eat until they were fully satiated. We did not measure the amount of food consumed during the devaluation procedure, only the effects of the devaluation on the hunger level (see our answer to comment 5 of reviewer 2) and the perceived pleasantness during the snack consumption before and after the devaluation manipulation.

Reviewer 1, minor comment 3: *Did pleasantness ratings differ between the two food identities? And if so, does this introduce a confound in the analysis?*

Authors’ response: Differences in the pleasantness of the two food outcomes would have indeed introduced a confound. However, we tried to directly address this aspect in the methodological procedure by selecting the most pleasant sweet and savory outcomes. Upon this suggestion, we also verified statistically that the experimental manipulation was successful. Specifically, we tested if the pleasantness ratings differ as function of the outcome identity (sweet or salty) in interaction with the experimental run (first, second or third). The analysis did not reveal any significant effect, and in particular, the analysis did not reveal statistical evidence favoring a difference in pleasantness between the sweet and salty outcome ($\beta = -0.270$, $SE = 0.294$, 95% CI = $[-0.847, 0.307]$, $p = 0.$, $BF_{10} = 0.168$; see Sup. Fig. 1). We have now added this control analysis in the supplementary materials of the revised manuscript.

Reviewer 1, minor comment 4: *There were 60 trials per conditioning run. How many trials for each of the five CSs and for each of the three*

Sup. Fig. 1 : Pleasantness of the sweet and salty snack. Mean pleasantness ratings of sweet and salty snacks over the three runs of the experiment ± 1 SEM adjusted for within participants designs.

deviant outcomes per CS? What were the deviant outcome for the neutral CS?

Authors' response: There were 10 per CS condition (we had 2 CS – conditions to have the same number of trials for the behavioral analysis on the gaze direction and the pupil dilation during the acquisition phase). 3 out of these 10 trials were deviant outcomes for a total of 15 deviant outcomes per run. Of these 15 deviant outcomes 8 were specific to the characteristics of the outcome with 4 deviant outcomes specific to the outcome location (e.g., common identity, but deviant location) and 4 deviant outcomes specific to the outcome identity (e.g., common location, but deviant identity). The remaining 7 deviant outcomes involved the neutral condition (e.g., CS- followed by one of the snack outcomes or CS+ followed by no outcome). We now added Table in the supplementary material that summarizes this information (see Table 1)

Reviewer 1, minor comment 5: *Figure 4: If I understand this plot correctly, correlations of state PEs with both behavioral measures (location and identity) are shown in panel A, whereas panel B shows correlations with reward PE. If this is correct, then please correct the main text on the bottom of page 13 to reflect this (this text currently suggests that correlations with state PE for identity and location effects are shown in panel A and B, respectively*

Table 1: Summary of the trials per condition in one run of the Pavlovian task

	Salty Outcome Left	Right	Sweet Outcome Left	Right	No Snack
CS+ salty left	7	1	1	0	1
CS+ salty right	1	7	0	1	1
CS+ sweet left	1	0	7	1	1
CS+ sweet right	0	1	1	7	1
CS-	0	1	1	1	7
CS-	1	1	1	0	7

Note. The experiment included three runs, the last run was administered under extinction.

Authors' response: Yes that is corrected, thank you. We rectified the text.

Reviewer 1, minor comment 6: *The authors are aware that the side decoding analysis is confounded by motor responses and eye movements. Please describe in the methods section how exactly the maps from the side decoding analysis were masked to remove signals related to motor responses and eye movements.*

Authors' response: We agree that this is an important information and we now added it to the revised manuscript. The relevant text of the revised method section now reads as follows (p. 42)

The map of the side decoding was masked to remove motor movement and residual eye movements. **The motor movement mask was created based on the GLM to define affective value and perceptual attributes. More specifically by the contrast: *left motor response > right motor response* and the contrast: *right motor response > left motor response*. Two clusters**

from motor areas (left and right) where extracted with a threshold of $p > 0.005$ and used as a mask. The residual eye movements mask was created based on the GLM to define identity and side. We used the second level t-map to decode side. Two clusters over from the residual activations from the eyeball (left and right) were extracted with a threshold of $p < 0.01$ and used as a mask. The thresholds of the masks were set to be more lenient than their respective analysis of interest (i.e., $p < 0.005$ for the univariate analysis at $p < 0.001$ and $p < 0.01$ for the multivariate analysis at $p < 0.005$).

Reviewer 1, minor comment 7: *Training and testing the classifier on data from the same runs (even when training and testing on different trials) may introduce biases. I assume the authors used this approach to maximize the amount of training data. However, it would be reassuring if they could show that the basic results hold up when training and testing the classifier on trials from different runs. Perhaps this control analysis could be performed in the ROIs from the original analysis.*

Authors' response: That is correct, we wanted to maximise the number of trials while controlling for the low-level perceptual properties of the stimuli, but also to control for learning effects (stimulus-outcome associations are likely much stronger in the last run compared with the first one). As a sanity check, we did run the suggested analysis in the ROI from the original analysis. We divided the data in three folds one per run. We trained the classifier to decode the CSs left vs. CSs right on two runs and then test on the remaining one in a 3-Folds cross-validation. We used the same procedure to train the classifier to decode the CSs sweet vs the CSs salty. Using this approach we could decode the identity in right intraparietal sulcus (IPS; $ACC = 0.526$, $SE = 0.007$, 95% CI = [0.511, 0.540], $p = 0.0008$, $BF_{10} = 39.028$); the post central gyrus (PCG; $ACC = 0.521$, $SE = 0.010$, 95% CI = [0.500, 0.542], $p = 0.049$, $BF_{10} = 1.223$) and the post central gyrus (PCL; $ACC = 0.525$, $SE = 0.010$, 95% CI = [0.503, 0.547], $p = 0.025$, $BF_{10} = 2.060$) but not in the right Inferior Frontal Gyrus (IFG; $ACC = 0.510$, $SE = 0.010$, 95% CI = [0.488, 0.532], $p = 0.330$, $BF_{10} = 0.308$), and we could decode the side in the ROI covering parts of the Cuneus ($ACC = 0.614$, $SE = 0.023$, 95% CI = [0.567, 0.662], $p < 0.001$, $BF_{10} = 763.17$), the the ROI covering parts

Sup. Fig. 2 : Classifier accuracies for the outcome identity (**A**) and the outcome side delivery (**B**) in the ROI from the main MVPA analysis which cover parts of the intraparietal sulcus (IPS), of the supra marginal gyrus (SMG), of the LOC = Lateral occipital complex (LOC), of the inferior frontal gyrus (IFG), of the superior temporal lobule and paracentral lobule (PCL) and parts of the post central gyrus (PCG).

of superior temporal lobule and intraparietal sulcus (IPS; $ACC = 0.541$, $SE = 0.009$, 95% CI = [0.521, 0.562], $p < 0.001$, $BF_{10} = 122.82$), in the ROI covering parts of right middle temporal gyrus and the lateral occipital cortex (LOC $ACC = 0.529$, $SE = 0.013$, 95% CI = [0.501, 0.557], $p = 0.041$, $BF_{10} = 1.414$) but not in the ROI covering parts of the left and right supra marginal gyrus (SMG $ACC = 0.509$, $SE = 0.006$, 95% CI = [0.495, 0.522], $p = 0.180$, $BF_{10} = 0.460$; see Fig. for Revisions). Even though this approach provides similar results to the main analysis, it is confounded by the low-level perceptual features of the CS stimuli as can be seen in the unusually high accuracy of the Cuneus ROI. We reported this control analysis in the supplementary material (pp. 3-4).

Reviewer 1, minor comment 8: *I was not clear what the authors meant by “concatenated” when describing the GLMs. First, I assumed that they appended data from all runs to estimate a single parameter estimate for a given condition for all runs. But this would not work for the analysis described on page 42, which separates responses pre- and post-devaluation. Do the authors mean that the GLM included data from multiple runs but each condition was still modeled with different regressors to obtain independent parameter estimates? If the latter is the case (which the default in SPM), I would suggest removing the term concatenation as it is typically used to refer to what is done using `spm_fmri_concatenate`.*

Authors’ response: thank you for catching this source of confusion. We did use the default in SPM, modeling data from multiple runs with each condition modeled with different regressors. We now re-phrased the method section removing the term "concatenation". The relevant text now reads as follows:

"The design matrix of the GLM contained the trials from the two learning runs" (p. 39)

"First, we used SPM to built a GLM using the unsmoothed functional EPI volumes. We entered all the trials of the three runs into one design matrix." (p. 40)

"The second GLM was built using SPM by entering the last learning run and the test run in a design matrix." (p. 43)

Reviewer 1, minor comment 9: *Please check the reference list for typos.*

Authors’ response: Thank you, we now doublechecked the reference list of the revised manuscript.

Reviewer 2

In this manuscript authors aimed to differentiate brain areas involved in learning and encoding associations that are sensitive to changes in the value of an outcome from those that are not sensitive to such changes. They find that, regions whose activity correlates with reward prediction errors are sensitive to outcome devaluation, challenging the assumption that reward prediction errors are exclusively model-free. Similarly, brain areas correlating with state prediction errors were found to be less sensitive to outcome devaluation, challenging the assumption that state prediction errors are model-based. Such findings appear noteworthy and of broad interest. The manuscript is clearly written, the methodology is sound and the conclusions are supported by the data. I only have a few minor suggestions that should be considered before publication.

Authors' response: Thank you very much for your comments. We greatly appreciate your constructive review and positive feedback; we have addressed the comments in detail below.

Reviewer 2, comment 1: *p. 9-10 "outcome devaluation induced changes": what about RTs, did outcome devaluation have any effect on RTs at Test?*

Authors' response: We have now run the analysis testing the outcome devaluation changes on the reaction times during the test session. We did not find statistical evidence for a devaluation effect. We added this result in the revised manuscript; the relevant text now reads as follows (p. 10).

Reaction times. We measured also reaction times to guess which video was being displayed behind the black patches during the test session following the CS associated with the valued outcome and the devalued outcome. We did not find evidence for a statistically significant difference between the CS valued and the CS devalued conditions ($t = 0.006$, $SE = 0.008$; 90% $CI = [-0.009, 0.021]$, $p = 0.459$, $BF_{10} = 0.440$).

For your information, we also plotted the effect of interest (see Fig. for Revisions). We did not add this plot in the revised manuscript, but we are willing to do so if you deem it relevant.

Fig. for Revisions : Reaction times during the test session. Normalized reaction times (RT) during the test session during which participants had to guess which outcome was hidden behind two black patches after the perception of the conditioned stimulus (CS) associated with the valued and the devalued outcome ± 1 SEM adjusted for within participants designs.

Reviewer 2, comment 2: *p. 28 "the video appeared on the left or right white frame", what appeared on the other frame? Fig. 1a seems to show that the background of the clip appears on the other frame, please add this to the description. Is this background also presented in "no outcome" trials (p. 29)? You should specify this*

Authors' response: Yes, that is correct. We added this description in the method section. The relevant text now reads as follows (p. 29):

"Each trial was composed of (a) a cue presented for 1.5 s to 4.5 s in either the upper or lower white frames; (b) an empty screen with only the background white frames presented for 3 s, and (c) a video of the experimenter's hand delivering their favorite snack into a small bag lasting 3 s. When the video appeared in either the left or the right white frame (see Fig 1), a picture depicting the small bag without any action was displayed on the opposite side of the screen. If no video was displayed, both sides displayed a picture of the small bag without any action. "

Reviewer 2, comment 3: *p. 28 please specify that participants needed to press a button corresponding to the side of outcome delivery, and that RTs*

were collected.

Authors' response: We added this important information in the method section as suggested. The relevant text now reads as follows (p. 30):

"Participants were instructed to focus on the cue and to try to predict what was going to happen next. They were instructed to move their eyes freely around the computer screen, but to focus their gaze on the fixation cross during the inter-trial interval. Participants were asked to press the left key when the food outcome appeared on the left side of the screen and the right key when the food outcome appeared on the right side of the screen as quickly and accurately as possible. They were informed that the key-pressing task was a measure of their sustained attention, independent of the cue-outcome contingencies. "

Reviewer 2, comment 4: *p. 29 "At the end of each run, there was a break to be taken outside the scanner during which participants received a bag containing the snacks they collected during the task, to consume." Did this quantity vary between subjects and/or runs and/or outcomes? How much was the variations between subjects and/or runs and/or outcomes? Could variations in the amount of snacks eaten during the conditioning session, e.g. if more snacks were eaten for one outcome than the other, then have an effect on devaluation and the subsequent test session?*

Authors' response: That is indeed a very important aspect. Variations in the amount of food consumed during learning could induce satiation processes, impacting the outcome's affective salience and thus the strength of Pavlovian conditioning. We tried to control for this aspect by having the same number of outcome videos displayed for each run. However, the different snacks used as an outcomes were selected based on the participant's preferences and varied between participants in terms of calories. We accounted for this difference by adapting correspondence between the amount of food consumed at the end of each session to the amount of calories of each snack. This correspondence was proportional and varied from 1:2 to 1:6 according to the amount of calories per individual piece of the snack selected by the participant. Since participants were asked not to eat for at least 6

hours prior to the experimental session, they were hungry (hunger was rated as 7.5 out of 10), so they all ate the entirety of the snacks they received between runs. We modified the method section of the revised manuscript to add the relevant information that was missing. The relevant text now reads as follow:

"[...] Participants were prescreened to ensure they were not dieting and they were asked not to eat for at least 6 h before the experimental session (but were allowed to drink water). (p. 27)"

"[...]At the end of each run, there was a break taken outside the scanner during which participants received a bag of snacks to consume, which they collected during the task. The amount of food to consume depended on the amount of calories per individual piece of the snack the participant selected. Each participant ate the entirety of the snacks they received during the learning session. (p. 31)"

Moreover, we also added the information about the hunger level in the supplementary material (see Sup. Fig. 4):

Sup. Fig. 4 : Hunger level during the experimental session. Mean level of hunger ratings over the three runs of the experimental session. The outcome devaluation procedure was administered after the acquisition phase (run 1 and 2) and before the test phase (run 3). Error bars represent ± 1 SEM, adjusted for within-participants designs. Double asterisks indicate statistically significant differences between conditions that survived multiple corrections.

Reviewer 2, comment 5: *p. 30 "They were also asked to press a key to indicate which one of the two black patches was obscuring the outcome delivery video."*

Authors' response: We have clarified that sentence, it now reads as follows (p. 31):

"They were also asked to press a key to guess under which of the two black patches the outcome delivery video was being displayed."

Reviewer #1 (Remarks to the Author):

The authors did an excellent job responding to my comments. I don't have any remaining points and congratulate the authors on a great paper!

Reviewer #2 (Remarks to the Author):

The authors addressed all my comments.